# MemStranding: Adversarial attacks on temporal graph neural networks

## Abstract

Temporal graph neural networks (TGNN) have achieved significant momentum in many real-world dynamic graph tasks. While this trend raises an urgent to study their robustness against adversarial attacks, developing an attack on TGNN is challenging due to the dynamic nature of their input dynamic graphs. On the one hand, subsequent graph changes after the attacks may diminish the impact of attacks on seen nodes. On the other hand, targeting future nodes, which are unseen during the attack, poses significant challenges due to missing knowledge about them. To tackle these unique challenges in attacking TGNNs, we propose a practical and effective adversarial attack framework, MemStranding, that leverages node memories in TGNN models to yield long-lasting and spreading adversarial noises in dynamic graphs. The MemStranding allows the attacker to inject noises into nodes' memory by adding fake nodes/edges at arbitrary timestamps. During future updates, the noises in nodes will persist with the support from their neighbors and be propagated to the future nodes by molding their memories into similar noisy states. The experimental results demonstrate that MemStranding can significantly decrease the TGNN models' performances in various tasks.

## 1 Introduction

Dynamic graphs are prevalent in real-world scenarios, spanning areas like social media (Kumar et al., 2018), knowledge graphs (Leblay & Chekol, 2018), autonomous systems (Leskovec et al., 2005), and traffic graphs (Pareja et al., 2020). Unlike static graphs, whose nodes and edges remain constant, dynamic graphs evolve over time, introducing challenging tasks like link prediction and node classification on dynamically changing nodes and edges. Driven by successes of Graph Neural Networks (GNNs)(Kipf & Welling, 2016; Hamilton et al., 2017; Veličković et al., 2017; Xu et al., 2018), Temporal Graph Neural Networks (TGNNs) have emerged as a popular solution for dynamic graph tasks(Trivedi et al., 2019; Kumar et al., 2019; Rossi et al., 2020; Zhang et al., 2023; You et al., 2022). TGNNs leverage memory vectors in nodes to track temporal history, leading to state-of-the-art results in numerous tasks. As such, there is a pressing need to study their robustness towards adversarial attacks, especially since such attacks have shown significant efficacy against traditional GNNs (Wang et al., 2018; Tao et al., 2021; Zügner et al., 2018; Zou et al., 2021; Zügner et al., 2020; Ma et al., 2020; Zang et al., 2020; Bojchevski & Günnemann, 2019; Sun et al., 2022; Li et al., 2022). For example, in social media such as REDDIT (Kumar et al., 2018) may employ TGNN to predict if a comment from a user to a post (as a node) should be banned based on his/her comment histories (as edges to the nodes representing Reddit posts). With subtle adversarial attacks to these TGNN models, malicious/fake messages can easily bypass this checking functionality.

By modifying the input graphs with imperceptible and subtle perturbations, adversarial attacks can inject noises into the node features in GNNs, making the models yield incorrect or adversary-expected results. However, conducting adversarial attacks on TGNNs is nontrivial due to the dynamic graphs' inherent dynamism. The reasons are two-fold: Firstly, the noises introduced by the attacks can quickly decay due to the future changes on the graphs and subsequent updates on node memories in TGNNs, as what we called *Noise Decaying* issue. Secondly, the attacker only knows the graphs up to the attack time; hence, the attacks can hardly affect unseen nodes/edges. We called this *Knowledge Missing* issue. To avoid such issues with existing attacks, the attackers must conduct adversarial attacks right before each change of the dynamic graphs. However, this is infeasible for two reasons: First, one cannot accurately predict when there will be an update, hence cannot decide

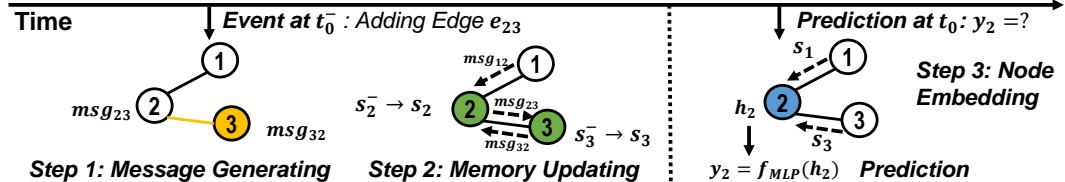

Figure 1: The three steps of TGNN computing assuming a new event at timestamp $t_0^-$ adds an edge $e_{23}$ to the dynamic graph: Firstly, messages are generated for the nodes that are directly involved in this event (i.e., $msg_{23}$ and $node_{32}$). Next, the nodes aggregate messages from their neighbors and update their memories (e.g., $s_2^- \rightarrow s_2$). At a future prediction time $t_0$, nodes aggregate memories (e.g., $s_1$ and $s_2$) from their neighbors and embed them into node vectors (e.g., $h_2$) for the prediction.

when to attack. Consequently, finding the attack time right before each change of the input graph is infeasible. Second, attacking before each update will introduce massive and frequent modifications on the input graph, making the attack evident and easy to defend. Therefore, an effective adversarial strategy against TGNN must withstand dynamic graphs' ever-evolving nature and influence current and future nodes without knowledge of future changes.

Fortunately, the memory updating mechanism in TGNNs exposes unique opportunities for persisting and propagating adversarial noises. In TGNN, a node's memory vector will be maintained to update its neighbors' memory vectors when these neighbors are involved in graph changes, such as node/edge adding/deleting. To this end, we raise an intriguing question:

> *Can we leverage the memory updating mechanism in TGNNs to persist and propagate adversarial noises injected at a specific timestamp?*

Drawing from these insights, we introduce MemStranding, a novel adversarial attack approach tailored for TGNNs. Given a specific attack timestamp, MemStranding strategically introduces fake nodes and edges to push sampled victim nodes' memories toward a stable, noise-infused state. These affected nodes retain the noisy memory and spread it to the future neighbors, aided by their connected but also affected nodes termed as *support neighbors*. Then, the affected future neighbors will also be converted to noisy states and contribute to persisting and spreading the noises in the future. Through this method, MemStranding effectively and persistently attacks both present and future nodes with limited knowledge up to the attack time. To the best of our knowledge, this paper presents the first adversarial attack on TGNNs.

## 2 BACKGROUND AND RELATED WORK

**Dynamic Graphs.** Unlike a static graph, which can be presented as a fixed set of nodes and edges as $\mathcal{G} = (V, E)$, a dynamic graph consists of nodes and edges evolving over time. Dynamic graphs can be represented in two ways: Discrete-Time Dynamic Graphs (DTDGs) describe dynamic graphs as a series of static snapshots taken periodically, while Continuous-Time Dynamic Graphs (CTDGs) view the graph as a collection of events—each event detailing updates like node or edge changes. Recent TGNNs focus on CTDGs since they can retain more information than DTDGs' fixed intervals and more complex (Kazemi et al., 2020) Within the CTDG paradigm, the dynamic graphs are represented as $\mathcal{G} = \{x(t_1), x(t_2), ...\}$, in which $x(t_i)$ indicates an event happened at timestamp $t_i$. Generally, the prediction task of deep learning models for CTDGs can be depicted in equation 1.

$$y_i = f_\theta(\mathcal{G}_i^-, t_i) = f_\theta(\{x(t_1), x(t_2), ...x(t_{i-1})\}, t_i) \tag{1}$$

At the prediction time $t_i$, the model $f_\theta(\cdot)$ takes all previous events $\mathcal{G}_i^- = \{x(t_1), x(t_2), ...x(t_{i-1})\}$ as inputs and predicts the testing nodes' classes or future edges.

**Temporal Graph Neural Networks.** The memory-based Temporal Graph Neural Networks (TGNN) are widely studied and achieve state-of-the-art accuracies in dynamic graph tasks (Trivedi et al., 2019; Kumar et al., 2019; Rossi et al., 2020; Kazemi et al., 2020; Zhang et al., 2023; You et al., 2022). Generally, these TGNNs maintain a node memory that tracks the node's history and uses it for the predictions. Here, we use the general framework presented in (Rossi et al., 2020) to introduce the workflow of TGNNs. As illustrated in Figure 1, TGNNs produce node embedding for

the predictions in three steps. When an event $x(t)$ adds an edge $e_{uv}$ from node $u$ to node $v$ (i.e., $x(t) = e_{uv}$), two messages are generated as equation 2.

$$m_{uv} = msg_s(s_u^-, s_v^-, \Delta T, e_{uv}), m_{vu} = msg_d(s_v^-, s_u^-, \Delta T, e_{uv}) \tag{2}$$

The $msg_s$ and $msg_d$ are learnable functions such as Multi-Layer-Perceptions (MLPs). The $s_u^-$ and $s_v^-$ denote the memories of node $u$ and node $v$ at their last updated times, and $\Delta T$ represents the difference between the current timestamp and the nodes' last updated times. Next, nodes $u$ and $v$ aggregate messages from their neighbors and update their memories as equation 3. For simplicity, we only present the updating and following operations of node $u$, which is the same for node $v$.

$$s_u^+ = UPDT(s_u^-, AGGR(m_{ku}^-|k \in N(u)), \tag{3}$$

The $N(u)$ denotes the neighbors of node $u$. The $AGGR(\cdot)$ is usually implemented by a $mean$ or $most\_recent$ function to aggregate messages from the node's neighbors (Rossi et al., 2020). The $UPDT(\cdot)$ uses the aggregated messages to update the node's memory and is usually implemented by a Gated-Recurrent-Unit (GRU) (Chung et al., 2014). When there is a prediction involving node $u$, TGNNs use a graph embedding module, such as Graph Attention Network (GAT) (Veličković et al., 2017), to embed the node's memory into the final node embedding, as depicted in equation 4.

$$h_u = GNN(s_u, s_k|k \in N(u)), \tag{4}$$

TGNNs use node memories from previous timestamps (i.e., $s_i$ and $s_u$) to compute the node embedding $h_u$ instead of using memories updated at the prediction timestamp to avoid information leakage. The resulting node embedding $h_u$ is fed into a final MLP module to get the prediction results. Generally, nodes relevant to the events at the current timestamp will be used as testing nodes (Rossi et al., 2020). For instance, if an event adds an edge $e_{uv}$ from $node_u$ to $node_v$ at timestamp $t$, then both nodes will be evaluated (e.g., predict nodes' classes or their outgoing edges) at timestamp $t$ based on their memories before updating.

**Adversarial Attacks on Graph Neural Networks.** The considerable achievements of GNNs have catalyzed numerous investigations into their resilience against adversarial attacks (Chen et al., 2017; Bai et al., 2018; Wang et al., 2018; Zügner et al., 2018; Bojchevski & Günnemann, 2019; Zügner et al., 2020; Ma et al., 2020; Zang et al., 2020; Tao et al., 2021; Zou et al., 2021; Sun et al., 2022; Li et al., 2022; Zou et al., 2023). These adversarial attacks generally seek to misguide GNN predictions by modifying the nodes and edges of input graphs. For example, (Wang et al., 2018) introduces fake nodes with fake features that can minimize the loss between prediction results in the original graphs and the targeted fake results; (Zügner et al., 2020) adds and deletes edges that can cause the most substantial increases in the training losses on the original graphs. Though different in their strategies, existing GNN attacks assume that (1) attackers have full knowledge of every node and edge in the targeted graphs and (2) these graphs remain static after the attack.

## 3 PROBLEM ANALYSIS

### 3.1 ATTACK MODEL

Adversarial attacks on TGNN face several constraints due to the inherent dynamism of the dynamic graphs. First and foremost, *attacks must be executed before the prediction's timestamp*. As outlined in equation 4, TGNNs generate node embeddings for the predictions from memories from previous timestamps, which are influenced by input events before the prediction, as depicted in equation 3. Hence, any adversarial attacks should occur before the prediction so that their updates can impact the resultant embeddings. Secondly, *attackers can only acquire knowledge up to the attack's timestamp*. At most, attackers can only access model details, presented inputs, and subsequent node memories but remain unaware of future changes in the dynamic graphs after the attack's timestamp. Thirdly, *they can add fake events as nodes/edges at the attack time*. For example, in social media, attackers can create fake user accounts as fake nodes and make junk comments to the blogs as fake edges.

### 3.2 CHALLENGES IN DYNAMIC GRAPHS

**Future interference.** Typical GNN attacks assume attackers have comprehensive knowledge of the target input graphs and that these graphs remain static after the attacks. However, in TGNNs, an

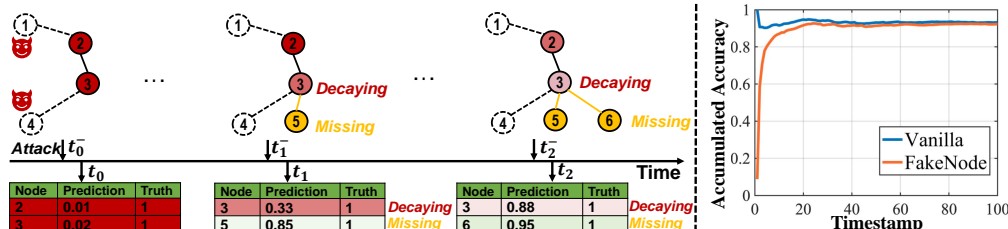

Figure 2: (Left) The noise decaying (*Decaying*) and knowledge missing (*Missing*) issues in TGNN adversarial attacks. Initially, at $t_0^-$, an attack adds fake $node_1$ and $node_4$, significantly skewing predictions for $node_2$ and $node_3$ at $t_0$. As time progresses to $t_1^-$ and $t_2^-$, the appearance of $node_5$, $node_6$, and subsequent updates dilute the adversarial impact on $node_3$, making its prediction closer to the ground truth. Moreover, the attacks can hardly work for these new nodes coming after $t_0^-$ (i.e., $node_5$, $node_6$) since the attackers lack knowledge of these nodes. (right) The accumulated accuracies on victim nodes in TGN under no-attack (vanilla) and FakeNode attack (FakeNode).

attacker's knowledge is limited to events up to the attack timestamp, and the graph continues to evolve after the attacks. To align with traditional attack assumptions, one might consider launching an attack immediately before each prediction; then, the attacker can tailor the attack based on full knowledge of the until-now inputs and ensure no subsequent changes before the prediction. However, this strategy is impractical for two reasons: Firstly, predicting the exact attack moment "just before" an unforeseeable future event is nearly impossible. Secondly, constantly modifying the graph before each prediction violates the imperceptible nature desired in adversarial attacks. To this end, attacks on TGNNs must endure interference from future changes in dynamic graphs.

**Issues from the future interference.** The inevitable future interference may potentially nullify the noise introduced by existing adversarial attacks towards GNNs. The reasons are two-fold: Firstly, for the seen and attacked targets, the noises that mislead their prediction would be mixed with information brought by the future changes (in equation 3 and equation 4), thereby no longer adequately strong enough to mislead their future predictions. We call this issue as *noise decaying*. Secondly, the unseen nodes and edges that were added after the timestamp of the attack can hardly be affected by the attacks. This is because the attackers have no idea about these future victims, hence unable to forge noises that can affect them. We call this issue as *knowledge missing*. We illustrate examples of noise decaying and knowledge missing in Figure 2. To further investigate the impact of the issues, we randomly sample 100 nodes from the Wikipedia dataset (Kumar et al., 2019), use the FakeNode attack (Wang et al., 2018) on a well-trained TGN (Rossi et al., 2020) and observe its performances. More details about the models and experiment setup are included in Section 5.1. As shown in Figure 2, while the attack effectively reduces the model's accuracy right after the attack timestamp, it can hardly perturb the predictions in the future.

**Opportunities in node memories.** While dynamic graphs considerably harden attacking TGNNs at specific timestamps, the intrinsic memory mechanism in these models offers pathways for more enduring and infectious adversarial attacks. Specifically, TGNNs maintain a memory vector for every node to store its historical data, which plays a pivotal role in all TGNN prediction stages as depicted from equations (2) to (4). This memory mechanism potentially facilitates noise transmission in three aspects: First, it can sustain attack-induced noises during updates as shown in equation 3; second, it can transmit these noises to neighboring nodes via the messages outlined in equation 2; and finally, it can influence neighboring nodes' embedding and subsequent predictions as described in equation 4.

## 4 THE MEMSTRANDING ATTACK

To ensure persistent and contagious adversarial noises in the dynamic settings of TGNNs, we introduce the MemStranding attack, which exploits the node memories in TGNNs to achieve the following goals: (1) Ensure that once a node's memory is contaminated, its adversarial noise remains sustained during updates, supported by its noisy neighboring nodes. (2) Ensure that a contaminated node's memory will influence its neighboring nodes' update processes, guiding their memory vectors toward a noisy state. MemStranding adds fake events to manipulate node memories, driving them into expected states. The resultant adversarial noises are enduring to the *noise decaying* issues and influence future nodes, addressing the *knowledge missing* issues, as illustrated in Figure 3.

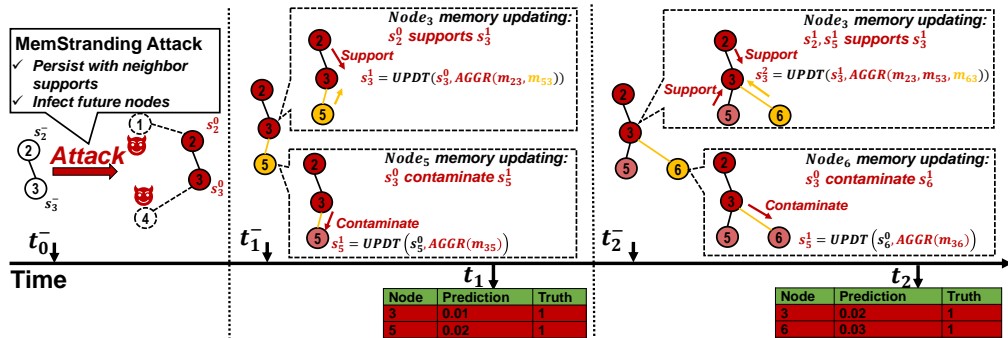

Figure 3: The overall goal of the MemStranding attack. (1) Persisting a node's noisy memory with support from its noisy neighbors, as $node_3$ updating at $t_1^-$ and $t_2^-$. (2) Contaminating a noisy node's new neighbors' memories, such as $node_3$ contaminating $node_5$ at $t_1^-$ and $node_6$ at $t_2^-$. As a result, despite any post-attack changes, all associated predictions could misguided.

## 4.1 NOISE PERSISTING

**Limits in Self-Persisting.** While nodes' memories show significant potential for retaining adversarial perturbations, achieving persistent noise in a node solely through its own memory, termed *self-persisting*, is challenging. Specifically, the unpredictable messages aggregated from its neighbors can influence a node's memory during its updating depicted in equation 3. An intuitive solution is to maximize the portion of the nodes' self-memory $s_u^-$ in the subsequent states while suppressing the effects of any incoming messages. We explored the viability of this approach with a case study in TGN (Rossi et al., 2020) by suppressing nodes' memory changes in its memory updating. We then assessed if nodes' memories remained consistent over time. As highlighted in Figure 4(a), this strategy fails to prevent future modifications. The primary challenge lies in the unpredictability of future messages, making it virtually impossible to find a memory that completely suppresses updates in the RNNs. Further experimental details and related theoretical proofs are provided in Appendix A.1. As a result, relying solely on *self-persisting* is insufficient to achieve our objectives.

**Opportunities in Cross-Persisting.** Although we cannot block messages from a node's neighbors, we can introduce noise to these messages, making them helpful for persisting noises as well. This approach, we termed as *cross-persisting*, is inspired by the graph smoothing phenomenon observed in GNNs (Li et al., 2018). The phenomenon suggests that when node features undergo repeated aggregations with neighboring features, they become over-smoothed and converge to similar values (Li et al., 2018). Drawing from this, we assume that nodes' noisy memories in TGN can remain consistent if they are surrounded by similar neighbors. We use the same model and data in the self-persisting experiment to verify the assumption. Specifically, for each sampled node (referred to as *root node*), we select two neighbors as its support nodes (referred to as *support neighbors*) and set their memories the same as the root node, then observe their memory changes over time. As depicted in Figure 4(b), with assistance from its neighbors, the memory within a node can quickly achieve a relatively stable state, persisting to future changes. Therefore, we propose to persist noise via *cross-persisting*. Specifically, we would first sample a node as the root node, then sample two of its neighbors as support nodes. Note that these support neighbors also cost our attack budgets, which limit the number of nodes to be attacked. Then, force them to fulfill the following two goals: First, the nodes' memories should be similar to each other so that their memories could have potential converge states. Second, the noisy nodes' memories should reach their converged states so that the process from the injecting states to the converged states (e.g., ascending part in Figure 4) will not distort the injected noises. While forcing one node's neighbors to be the same as itself may make its neighbors away from their own converge states, as shown in Figure 4(c), the converge states from different nodes are similar due to the graph smoothing effects. Hence, for each attacked node, we formulate the problem to be solved as equation 5.

$$\mathcal{L}_u^{persist} = \sum_{k \in N^*(u)} \left( \mathcal{L}_{mse}(s_k^*, s_k^+) + \mathcal{L}_{mse}(s_u^+, s_k^+) \right) \qquad (5)$$

For any given $node_u$ with its memory denoted as $s_u$ and support nodes as $N^*(u)$, our objective relies on two Mean-Squared-Error (MSE) losses. The first, $\mathcal{L}_{mse}(s_k^*, s_k^+)$, aims to ensure that it updates its support neighbors' memory $s_k^+$ close to their ideal or converged state $s_k^*$. We introduce

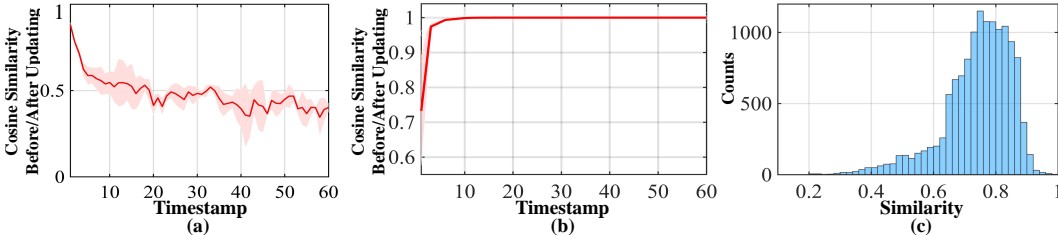

Figure 4: The ranges (colored bar) and averages (line) of the cosine similarities between node's evolving memories over time with (a) *self-persisting* and (b) *cross-persisting* solutions. (c) The distribution of cosine similarities between the converged states in different nodes.

how to find $s_k^*$ in Section 4.3. The second loss, represented by $\mathcal{L}_{mse}(s_u^+, s_k^+)$, is designed to make sure that its own updated memory $s_u^+$ are similar to the updated memories $s_k^+$ of its support nodes. By leveraging these losses, we aim to drive the nodes' memory into a state that strikes a balance between its own and neighbors' converged states.

## 4.2 NOISE PROPAGATING

**Contamination exercise.** To make the nodes' memory influential for future neighbors, we propose to use its existing neighbors to simulate those potential future neighbors. This is because, due to the principle of homophily in real-world graphs, neighboring nodes often exhibit close similarities (McPherson et al., 2001). Specifically, for a given $node_u$, we first augment its current neighbor set $N(u)$ with several "fake future neighbors", which are created by sampling nodes from $N(u)$ and mixing Gaussian noises to them. We then use the resulting augmented neighbors $N'(u)$ to solve the problem described in equation 6.

$$\mathcal{L}_u^{prop} = \sum_{k \in N'(u)} \mathcal{L}_{mse}(s_u, UPDT(s_k, m_{uk})) \tag{6}$$

The objective of this loss function is to minimize the Mean Squared Errors (MSEs) between a node's memory and the memories of its new neighbors after an update. By implementing this strategy, we ensure the memory becomes contagious for its current and potential future neighbors.

**Neighbor-aware Cross-optimization.** So far, for a node $u$, we can combine its losses for noise persisting and propagating as equation 7.

$$\mathcal{L}_u = \mathcal{L}_u^{persist} + \mathcal{L}_u^{prop} \tag{7}$$

We utilize the Adam optimizer (Kingma & Ba, 2014) to optimize the nodes' memories based on their gradients. To bolster the nodes' propagation capabilities, we employ a cross-optimization strategy. This strategy calculates the gradients of one node's memory relative to its support neighbor's loss and uses these gradients to solve the node's memory in each optimizing step, as outlined in equation 8.

$$s_u^{q+1} = s_u^q - \alpha \cdot \nabla_u \mathcal{L}_k \tag{8}$$

In this equation, $q$ indicates the optimizing steps, and $\alpha$ denotes the learning rate. Intuitively, the cross-optimization scheme encourages a node's memory to support its neighbors in achieving the optimization goals. By doing so, the solved node memories can better forge their neighbors and vice versa, thereby further improving their capacity to contaminate others.

## 4.3 ATTACK FRAMEWORK

Combining the above-mentioned solutions together, we introduce an adversarial attack framework, MemStranding, that conducts the attack on TGNs in two stages, as illustrated in Figure 5.

**Stage 1: Victim Node Sampling.** In this stage, we use a simple greedy approach to select the nodes to be attacked. Specifically, we sample the nodes with the highest degrees in the current graph as root nodes. The intuition behind this is that we want the injected noises to be propagated to as many nodes as possible, and these high-degree nodes, such as top-commented posts on social media, are usually popular in existing and future graphs. Next, for each root node, we sample its two highest-degree neighbors as its support nodes. All the root and support nodes will be treated as victim nodes and have their noise solved and affected in the following procedure.

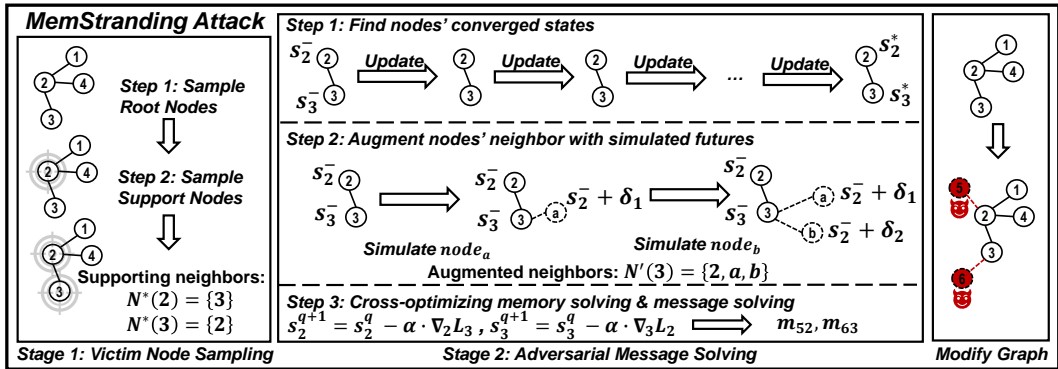

Figure 5: The two stages of the MemStranding attack. In the **victim node sampling stage**, we greedily sample victim nodes under the attack budget. In the **adversarial message solving stage**, we solve messages to be injected to achieve our attack goal. The solved messages are added to the graphs and removed after the attack timestamp. We included a detailed algorithm in Appendix A.2.

**Stage 2: Adversarial Message Solving.** In this stage, we solve the messages to be passed to the sampled victim nodes, which can be injected as fake nodes or edges. In the first step, we find the nodes' converged states (i.e., $s_u^*$ in equation 5) by updating its memory using current neighbors until convergence. In the second step, we simulate the future changes by augmenting victim nodes' neighbors with simulated futures (i.e., nodes/edges). The resulting neighbors are used as $N^+(u)$ in equation 6. In the third step, we use *cross-optimization* following equation 8 to find the expected noisy memories $\hat{s}_u$ of the victim nodes. Then, we solve the adversarial messages described in equation 9 so that the nodes' memory can reach these expected states after injecting them.

$$\underset{m_{\mathcal{A}u}}{\arg\min} \mathcal{L}_{mse}(UPDT(s_u^-, AGGR(m_{\mathcal{A}u}, \widetilde{m_u})), \hat{s}_u) \tag{9}$$

In short, for $node_u$, the solution aims to find a fake message $m_{\mathcal{A}u}$ that minimizes the MSE loss between the expected noise memory $\hat{s}_u$ and the memory updated after inserting it to the graph. The $\widetilde{m_u}$ represents the aggregated messages collected from the rest of $node_u$'s neighbors. Lastly, we can add the solved noisy message as a fake node or fake edge accordingly and remove it after the attack.

## 5 EVALUATION

### 5.1 EXPERIMENTAL SETUP

**Models and Datasets:** We use on four TGNN models for evaluation: Jodie (Kumar et al., 2019), Dyrep (Trivedi et al., 2019), TGN (Rossi et al., 2020) and Roland (You et al., 2022). The experiments use four dynamic graph datasets: Wikipedia (`WIKI`), Reddit (`REDDIT`) (Kumar et al., 2019), Reddit-body (`REDDIT-BODY`) and Reddit-title (`REDDIT-BODY`) (Kumar et al., 2018). More details about the models and datasets are included in Appendix B.1.

**Training Task & Metrics:** We train and evaluate the models on tasks: node classification and edge prediction (Rossi et al., 2020). We use the commonly used Area under the ROC Curve (ROC-AUC) to measure the model performances. For a timestamp, we measure the accuracies/AUC based on all presented predictions from the beginning, which we termed as *accumulated accuracy* and *accumulated ROC-AUC*. More details about the tasks and matrices in Appendix B.2

**Attack Setup:** We compare our work with three state-of-the-art GNN attacks: FakeNode (`FN`) (Wang et al., 2018), TDGIA(`TDGIA`) (Zou et al., 2021) and Meta-Attack-Heuristic(`Meta-h`) (Li et al., 2022). The results from Table 1 evaluate all attacks with 5% attack budgets, where we inject noises to 5% nodes of the input graph. In Appendix B.3, we evaluate attacks with 1% attack budgets. For our attack, we use a $1/3$ budget for root node sampling and $2/3$ for support nodes.

**Defense Setup:** We adopt two typical adversarial defenses–Adversarial Training(`Adv_train`) and— Regularization under empirical Lipschitz (`Lip_reg`)–from static GNNs to train robust TGNs, and evaluate the proposed attack on these robust models. Details are in Appendix B.4.

Table 1: Accumulated accuracy of edge prediction in the vanilla/attacked TGNNs over different timestamps; lower matrices indicate more effective attacks. More results in Appendix B.3

| Dataset | | WIKI | | | | REDDIT | | | | REDDIT-BODY | | | |
|---|---|---|---|---|---|---|---|---|---|---|---|---|---|
| **Model** | | TGN | Jodie | Dyrep | Roland | TGN | Jodie | Dyrep | Roland | TGN | Jodie | Dyrep | Roland |
| Vanilla | | 0.93 | 0.87 | 0.85 | 0.94 | 0.96 | 0.98 | 0.96 | 0.95 | 0.90 | 0.87 | 0.90 | 0.88 |
| $t_0$ | FN | 0.81 | 0.74 | 0.74 | 0.82 | 0.84 | 0.83 | 0.84 | 0.83 | 0.76 | 0.82 | 0.77 | 0.79 |
| | Meta-h | 0.90 | 0.83 | 0.81 | 0.85 | 0.93 | 0.95 | 0.90 | 0.92 | 0.86 | 0.83 | 0.88 | 0.85 |
| | TDGIA | **0.77** | **0.72** | **0.71** | **0.80** | **0.74** | **0.80** | **0.81** | **0.74** | **0.72** | **0.81** | **0.74** | **0.76** |
| | Ours | 0.89 | 0.88 | 0.85 | 0.87 | 0.75 | 0.84 | 0.94 | 0.82 | 0.84 | 0.85 | 0.81 | 0.78 |
| $t_{25}$ | FN | 0.92 | 0.87 | 0.85 | 0.94 | 0.97 | 0.97 | 0.96 | 0.93 | 0.90 | 0.86 | 0.89 | 0.88 |
| | Meta-h | 0.93 | 0.87 | 0.84 | 0.93 | 0.96 | 0.98 | 0.94 | 0.96 | 0.89 | 0.86 | 0.90 | 0.87 |
| | TDGIA | 0.93 | 0.81 | 0.84 | 0.92 | 0.94 | 0.95 | 0.95 | 0.90 | 0.89 | 0.85 | 0.89 | 0.88 |
| | Ours | **0.80** | **0.80** | **0.78** | **0.85** | **0.81** | **0.84** | **0.91** | **0.80** | **0.81** | **0.84** | **0.76** | **0.80** |
| $t_{50}$ | FN | 0.94 | 0.87 | 0.86 | 0.94 | 0.97 | 0.97 | 0.96 | 0.95 | 0.90 | 0.86 | 0.90 | 0.88 |
| | Meta-h | 0.93 | 0.87 | 0.85 | 0.93 | 0.97 | 0.98 | 0.94 | 0.95 | 0.90 | 0.86 | 0.90 | 0.88 |
| | TDGIA | 0.93 | 0.87 | 0.85 | 0.93 | 0.96 | 0.97 | 0.95 | 0.92 | 0.89 | 0.86 | 0.90 | 0.87 |
| | Ours | **0.75** | **0.81** | **0.76** | **0.84** | **0.80** | **0.84** | **0.91** | **0.80** | **0.77** | **0.82** | **0.76** | **0.77** |

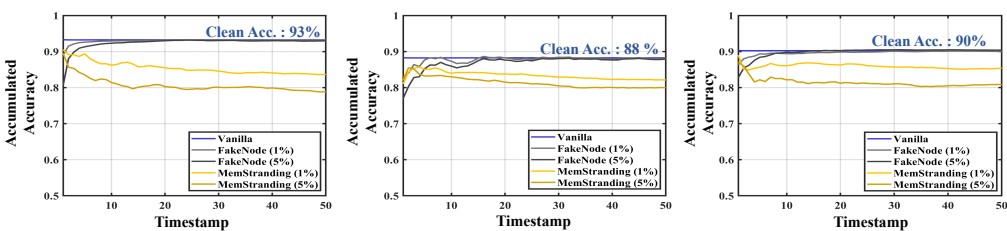

Figure 6: Accumulated accuracies of TGN under no defense(left), `Adv_train`(middle), and `Lip_reg`(right) with FakeNode and our attack on `WIKI` dataset. More results in Appendix B.4

## 5.2 EXPERIMENTAL RESULT

**Overall Performance**. We collect the accumulated accuracy at three timestamps: $t_0 = 0$, $t_{25} = 25$, and $t_{50} = 50$, which indicate time intervals after the attack noises are injected into the memories. The results of the edge prediction task are shown in Table 1, and more results of the edge prediction and node classification tasks are included in Appendix B.3 As one can observe, all prior attacks achieve significant accuracy drops at the $t_0$ timestamp, but the strengths quickly decay as time shifts. In $t_{25}$ and $t_{50}$, the accumulated ACC under these attacks is nearly identical to the vanilla. In contrast, our proposed MemStranding fabricates the noise with persistence. Since MemStranding spends more budget on injecting the noise to the support nodes, it does not achieve considerable accuracy drops at $t_0$ compared to the FakeNode attack. Nevertheless, as the time shifts to $t_{25}$ and $t_{50}$, MemStranding maintains a high accuracy drop over the entire time interval. Moreover, when the new event arrives, MemStranding propagates its noise to new incoming neighbors and affects more nodes. As a result, the accuracy drop increases at the later timestamps. It is worth mentioning that the attacks are weak on Jodie since it adopts a memory decay mechanism that uniformly decays all previous memories. Besides, memory decay delivers additional information outside the node memory, making Jodie more robust toward attacks from node memories. We further illustrate the accumulated accuracy of TGNs under different defenses in Figure 16. The typical static GNN attack (i.e., FakeNode) fails to perturb the model over time, i.e., when $t = 50$, the FakeNode attack only achieves 1.1% accuracy drop. In contrast, under the MemStrending attacks, the model suffers continually accumulated accuracy drops.

**Ablation Studies.** To analyze the propagating and persisting capability of the noise solved by MemStranding, we capture 100 root nodes in TGN in edge prediction on `WIKI` and monitor the changes in their memory and their neighbors' memory. In Figure 22, we compare the cosine similarity between the memories of the root nodes at $t_0$ with those in themselves and their one-hop and two-hop neighbors at each timestamp after the attack in four versions: (1) MemStranding, (2) MemStranding w/o converge state (i.e., without starting from converge state and using $\left( \mathcal{L}_{mse}(s_k^*, s_k^+) \right.$ in Equation equation 5), (2) MemStranding w/o persisting loss (i.e., without using $\mathcal{L}_u^{persist}$ Equation equation 5), and (4)those similarities in the original TGNN without attacks. The result shows that the noise in the root node can persist over 10 timestamps, with over 0.92 cosine similarities. For

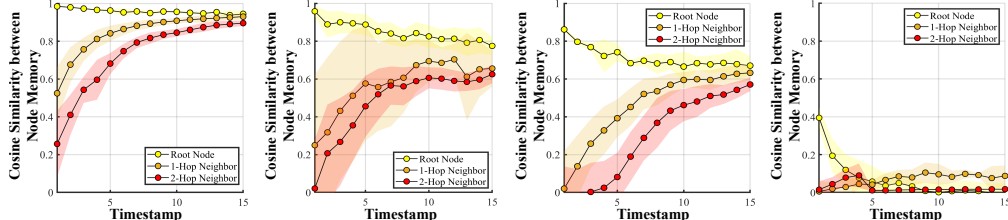

Figure 7: The similarities between root nodes' initial noisy memories (at the time of the attack) and themselves'/their subsequent neighbors' memories in MemStranding(left), MemSranding w/o (middle-left) converge state, MemSranding w/o persisting loss (middle-right), and regular nodes (right). All results above are from TGN and `WIKI`. More results in Appendix B.5.

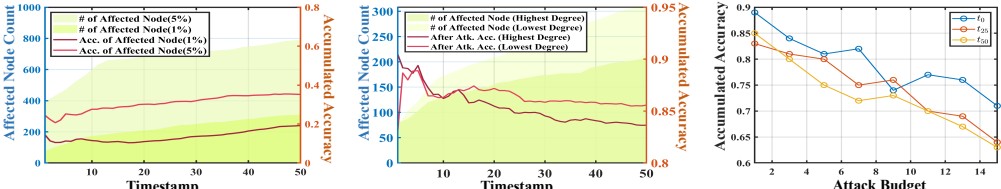

Figure 8: (left) Count of nodes affected by MemStranding and accuracy for the affected nodes over time. (middle) Comparison between two strategies for selecting the injected node: 1% lowest degree and highest degree nodes. Count of affected nodes and overall accuracy over time. (right) The accumulated accuracy at $t_0$, $t_{25}$, and $t_{50}$ under different attack budgets (% of total nodes). All results above are from TGN and `WIKI`. More results in Appendix B.6.

the one-hot neighbors, at $t = 1$, they achieve 0.51 average similarities after the first update by the message from root nodes, and at $t = 15$, the average rises to 0.88. For the two-hot neighbors, whose memories are updated by the message from one-hot neighbors, their average similarities grow from 0.24 to 0.84. In contrast, the similarity between nodes' initial attacked memory and their future counterparts drops drastically in the normal TGNNs and fails to propagate to their neighbors and persist them. If the converge states are not guaranteed, the similarities also suffer drops and achieve much lower similarities in the future. This is because the memories will change before they reach their converged states, making the final converged state different from the original noise ones, indicating that the converged state is essential for persisting noisy memories. The similarities drop faster if we remove the persisting loss since the cross-persist is entirely disabled. Moreover, despite the removal of persisting losses, the neighbors are getting closer to the target nodes, indicating that the propagating loss works as expected. The results demonstrate that the noises solved by the MemStranding can effectively propagate to their neighbors and boost stronger later.

**Sensitive Studies.** In Figure 5.2 (left), We evaluate MemStranding by its affected nodes (presented as the colored area). Specifically, we monitor all the topologically connected nodes to the root node after the noise is injected. Further, we measure the prediction accuracy of the affected nodes (presented as colored lines). The results indicate that MemStranding can effectively persist noises in affected nodes despite their future changes. In Figure 5.2 (middle), we evaluate two strategies for selecting the nodes for the noise injection: selecting the nodes with the top 5% highest degree and the nodes with top 5% lowest degree. MemStranding employs the former approach, resulting in rapid noise propagation and greater accuracy degradation. In Figure 5.2 (right), we evaluate MemStranding in attack budgets ranging from 1% to 15%. With more attack budget, MemStranding achieves a higher accumulated accuracy drop.

## 6   CONCLUSION

In this work, we investigate the challenges and opportunities in adversarial attacks on memory-based TGNNs. Based on our observations, we propose MemStranding, a novel adversarial attack tailored for TGNNs, to overcome the challenges brought by the dynamism in TGNNs. Firstly, it utilizes memories from a node's neighbors to persist adversarial noises over time. Secondly, it molds a node's memory so that the node can affect future neighbors by polluting them into a similar noisy state. The experimental results show that our approach can produce long-lasting and contagious noises over a long period in the dynamic graphs, leading to significant performance drops in TGNNs.

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

## APPENDIX

## A EXTENDED DESIGN

### A.1 SELF-PERSISTING EXPERIMENTAL SETUP

We explored the viability of the self-persist approach with a case study in TGN (Rossi et al., 2020), where the $UPDT(\cdot)$ function is typically realized using a GRU (Chung et al., 2014). At a particular timestamp, we randomly sample 100 nodes from the Wikipedia dataset and modify their memories. For each node, we use Adam optimizer (Kingma & Ba, 2014) to find a memory vector to suppress GRU updates by minimizing its reset gates (Chung et al., 2014). We then assessed if this memory state remains consistent over time.

The TGN used by the self-persisting experiment uses GRU for memory updating (i.e., for implementing $UPDT(\cdot)$ function in equation 3), as depicted in equation 10-13.

$$r_t = \sigma(W_{ir}\widetilde{m_t} + b_{ir} + W_{hr}s_{t-1} + b_{hr}) \tag{10}$$

$$z_t = \sigma(W_{iz}\widetilde{m_t} + b_{iz} + W_{hz}s_{t-1} + b_{hz}) \tag{11}$$

$$n_t = tanh(W_{in}\widetilde{m_t} + b_{in} + r_t \odot \sigma(W_{in}\widetilde{m_t} + b_{in})) \tag{12}$$

$$s_t = (1 - z_t) \odot n_t + z_t \odot s_{t-1} \tag{13}$$

where $\sigma(\cdot)$ is the sigmoid function. Given the node memory $s_{t-1} \in \mathbb{R}^M$ at the previous timestamp, and the aggregated message $\widetilde{m_t} \in \mathbb{R}^D$ at time $t$, GRUs first compute reset gate $r_t \in \mathbb{R}^M$, update gate $z_t \in \mathbb{R}^M$, and new gate $n_t \in \mathbb{R}^M$.

In the self-persisting experiment, we aim to minimize the interference of the message, $\widetilde{m_t}$, and maintain the updated memory, $s_t$, close to the previous memory, $s_{t-1}$. To this scope, we can maximize all the features in the update gate, $z_t$, until it approaches $\mathbb{1}$, where the update gate will be directly used to control the portion of the previous memory, which is:

$$\text{as } z_t \to \mathbb{1}, \quad s_t \to \mathbb{0} \odot n_t + \mathbb{1} \odot s_{t-1} \approx s_{t-1} \tag{14}$$

Additionally, according to Equation 11, the update gate $z_t$ is computed by the sum of two linear processes, and one is from the message, $\widetilde{m_t}$ and the other one is from memory $s_{t-1}$. As we maximize the linear output of the memory, $W_{hz} \cdot s_{t-1}$, the update gate, $z_t$, is then maximized.

Hence, to analyze the maximum output of the linear process, $W_{hz} \cdot s_{t-1}$, we formulate it into a linear program problem with the equations:

$$\begin{aligned} \max \quad & \sum W_{hz} \cdot s_{t-1} \\ \text{s.t.} \quad & -\mathbb{1} \le s_{t-1} \le \mathbb{1} \\ & W_{hz} \cdot s_{t-1} > \delta \end{aligned} \tag{15}$$

As the memory is the output of the $tanh$ function rather than the unit-length vector, $s_t$ is bounded by the limit of the $tanh$ function, $[-1, 1]^M$. Further, we introduce an addition constraint $W_{hz} \cdot s_{t-1} > \delta$ to guarantee all dimensions of the linear output are bound by a constant, $\delta$.

The optimal result for the memory, $s_{t-1}^*$, for the linear problem only depends on the model weights, where given a TGN model, the solution of the self-persisting memory is unique, and we have conducted the experiment on three models TGN+WIKI, TGN+REDDIT, and a randomly initialized model.

The result in Figure 9-11 (a) shows the maximum update gate, $z_t^*$, computed by $\sigma(W_{hz} \cdot s_{t-1}^*)$. In the TGN+wiki example, $z_t^*$ is a 172-dimension vector, and it is distributed with a mean of 0.64 and a standard deviation of 0.12. As aforementioned, to achieve the self-persisting memory, the update gate, $z_t$, is required to approach $\mathbb{1}$, but it is infeasible to fine the solution in the real world case under the constraints. In Figure 9-11 (b), we simulate the GRU updating starting with the optimal memory, $s_{t-1}^*$, and monitor the cosine similarity between the memory before updated and after updated. The results further demonstrate even the optimal solution cannot accomplish the self-persisting goal.

To theoretically analyze the maximum of the in the general case, we divide them into their eigenrepresentations, and we use the SVD decomposition:

$$W_{hz} = U \cdot \Sigma \cdot V^T = \sum_i e_i \cdot U_i \cdot V_i^T, \quad s_{t-1} = \sum_{i|V_i \in V} \alpha_i \cdot V_i \tag{16}$$

In SVD decomposition, $U$ and $V$ are the unitary matrix, and we use the basis from $V$ to decompose $s_{t-1}$. Moreover, the linear process is written as:

$$W_{hz} \cdot s_{t-1} = \sum_i e_i \cdot \alpha_i \cdot U_i \cdot V_i^T \cdot V_i = \sum_i \alpha_i \cdot e_i \cdot U_i \qquad (17)$$

This linear process is represented by the linear combination on the basis of $U$. We can easily acquire the theoretical maximum of the output. As $s_{t-1} \in [-1, 1]^M$, if $V_\theta$ is a basis of $\{-\frac{1}{\sqrt{M}}, \frac{1}{\sqrt{M}}\}^M$, $s_{t-1} = s_\theta$ can achieve the maximum projection to this basis, which is, $\alpha_\theta = \sqrt{M}$ and $s_{t-1} = V_\theta$. Similarly, the linear output $W_{hz} \cdot s_{t-1}$ achieves the maximum by there exist a basis $U_\theta = \{\frac{1}{\sqrt{M}}\}^M$, and the linear output,

$$W_\theta \cdot s_\theta = e_\theta \cdot \frac{1}{\sqrt{M}} \cdot \sqrt{M} \cdot \mathbb{1} = e_\theta \cdot \mathbb{1} \qquad (18)$$

As is shown, the maximum output of the linear process is equal to the eigenvalue. According to the experiment, the largest eigenvalue of the weight matrix is usually around 2. Therefore, the update gate, $z_t$, has the theoretical maximum value, $\sigma(e_\theta) \approx 0.88$.

However, the weights, $W_{hz}$, are trained through the model update, which makes it impossible to find the ideal maximum in the practical case.

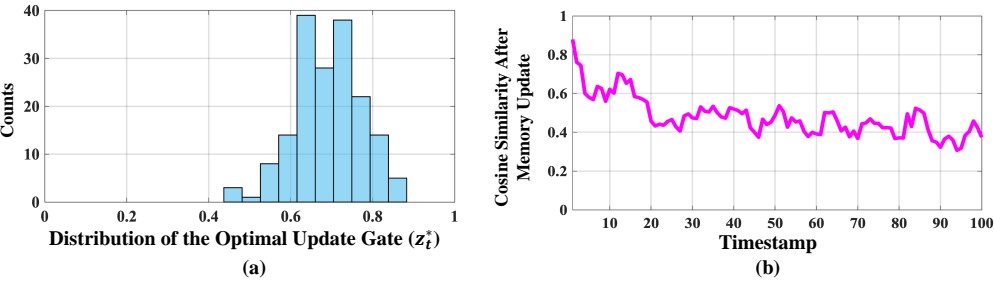

Figure 9: (a) The distribution of the optimal update gate $z_t^*$. (b) The cosine similarity between memory before the update and after the update, starting with the optimal self-persisting memory $s_t^*$. Experiments are conducted in the TGN model with `WIKI` datasets.

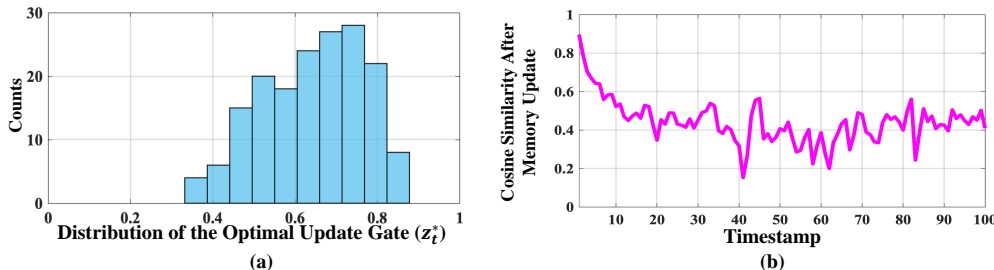

Figure 10: (a) The distribution of the optimal update gate $z_t^*$. (b) The cosine similarity between memory before the update and after the update, starting with the optimal self-persisting memory $s_t^*$. Experiments are conducted in the TGN model with `REDDIT` datasets.

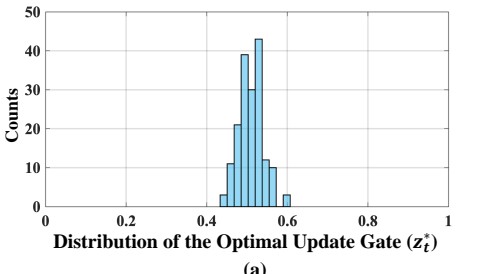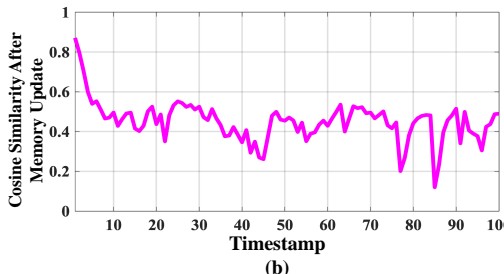

Figure 11: (a) The distribution of the optimal update gate $z_t^*$. (b) The cosine similarity between memory before the update and after the update, starting with the optimal self-persisting memory $s_t^*$. Experiments are conducted in the randomly sampled GRU model.

## A.2   Design: Overall Algorithm

We present the detailed algorithm of our approach in this section.

---

**Algorithm 1 MemStranding Attack**

---

**Input**  : $\mathcal{G} = (V, s(V)) \leftarrow$ Original graph with Node $V$, memories $s(V)$
**Input**  : $\forall_{i,j|V_i, V_j \in V} \quad m(s_i, s_j, e_{ij}, \Delta t) \leftarrow$ Messages before $t_0$
**Input**  : $\mathcal{B} \leftarrow$ Number of attacked nodes (attack budget)
**Input**  : $q \leftarrow$ Number of support neighbors for each root node.
**Output:** $V_{\mathcal{A}}, e_{\mathcal{A}}$: Perturbed nodes and message.

/* **Stage 1. Victim Node Sampling** */
$n \leftarrow \mathcal{B}/(q+1)$
**for** $i \in \{1, 2, \cdots, n\}$ **do**
   $V_i^{\text{root}} \leftarrow topk(\text{degree}(V), n)$
   $V_{i,1:q}^{\text{support}} \leftarrow V_i^{\text{root}} \cup \{V_1, V_2, \cdots V_q \in N(V_i^{\text{root}})\}$
   $\hat{s}_{i,1:q} \leftarrow \texttt{ComputeConvergeState}(V_i^{support})$
   $V_{\mathcal{A}}, e_{\mathcal{A}} \leftarrow \texttt{ComputeAdversarialMessage}(\hat{s}_{i,1:q}, V_{i,1:q}^{support})$

/* **Stage 2. Solving Converge State** */
**Function** $\texttt{ComputeConvergeState}(V^{support})$
   /* **2.1. Solving the self-optimal State** */
   **for** $V_i \in V^{support}$ **do**
      $s_i \leftarrow s(V_i)$
      $m \leftarrow m(s(V_i), s(V_i), e_{ij}, \Delta t) \mid V_j \in V$
      **do**
         $s_i \leftarrow s_i^+$
         $m \leftarrow m(s_i, s_i, e_{ij}, \Delta t)$
         $s_i^+ \leftarrow UPDT(s_i, m)$
      **while** $||s_i^+ - s_i||_2^2 > \epsilon$;
   $s_1^*, s_2^*, \cdots s_q^* \leftarrow s_1^+, s_2^+, \cdots s_q^+$

   /* **2.2. Solving the cross-optimal State** */
   $s_1^{(0)}, s_1^{(0)}, \cdots s_q^{(0)} \leftarrow s_1^*, s_2^*, \cdots s_q^* + \mathcal{N}(0, \eta \cdot \sigma(s(V)))$
   **for** $t \in \{0, 1, 2, \cdots, T\}$ **do**
      $\forall_{i \in \{1,2,\cdots q\}}, \quad s_i^{(t)+} \leftarrow UPDT(s_i^{(t)}, \widetilde{m_i})$
      **for** $i \in \{0, 1, 2, \cdots, q\}$ **do**
         $\mathcal{L}_i^{persist} \leftarrow \sum_{k \in N^*(i)} \left( \mathcal{L}_{mse}(s_k^{(t)+}, s_k^*) + \mathcal{L}_{mse}(s_i^{(t)+}, s_k^{(t)+}) \right)$
         $\mathcal{L}_i^{prop} \leftarrow \sum_{k \in N'(u)} \mathcal{L}_{mse}(s_i^{(t)}, UPDT(s_i^{(t)}, m_{ik}))$
      $\forall_{i \in \{1,2,\cdots q\}}, \quad s_i^{(t+1)} \leftarrow s_i^{(t)+} - \alpha \cdot \nabla_{s_i}(\mathcal{L}_k^{persist} + \mathcal{L}_i^{prop})$
   **return** $\{s_1^{(T)}, s_2^{(T)}, \cdots s_q^{(T)}\}$

/* **Stage 3. Solving the Adversarial Message** */
**Function** $\texttt{ComputeAdversarialMessage}(\hat{s}, V^{support})$
   **for** $V_i \in V^{support}$ **do**
      $V_{i,\mathcal{A}} \leftarrow V \in N(V_i)$
      **for** $t \in \{0, 1, \cdots, T\}$ **do**
         $m_{\mathcal{A}i}^{(t)} \leftarrow m(s(V_i), s(V_{\mathcal{A}}), e_{\mathcal{A}i}^{(t)}, \Delta t)$
         $\mathcal{L}_{\mathcal{A}} \leftarrow \mathcal{L}_{mse}(UPDT(s(V_i), AGGR(m_{\mathcal{A}i}^{(t)}, \widetilde{m_i}), \hat{s}_i)$
         $e_{\mathcal{A}i}^{(t+1)} \leftarrow e_{\mathcal{A}i}^{(t)} - \alpha \cdot \nabla_{e_{\mathcal{A}i}^{(t)}} \mathcal{L}_{\mathcal{A}}$
   **return** $\{V_{1,\mathcal{A}}, V_{2,\mathcal{A}}, \cdots, V_{q,\mathcal{A}}\}, \{e_{\mathcal{A}1}, e_{\mathcal{A}2}, \cdots, e_{\mathcal{A}q}\}$

---

# B EXTENDED EVALUATION

## B.1 EXPERIMENTAL DETAILS.

**Model Details**. All four TGNNs we included maintain a memory vector in each node and follow the memory updating process as discussed in Section 2. And they are different in their node embedding procedure (i.e., equation 4). Specifically, Dyrep directly uses the node memories for the predictions (i.e., $h_i^t = s_i^t$). Jodie applies a time-decay coefficient to the scale memories before classification (i.e., $h_i^t = \delta(t) \cdot s_i^t$). TGN, on the other hand, refines memories using a single-layer graph attention module, as outlined in equation 4. Unlike prior models, ROLAND You et al. (2022) is a recent model designed for DTDG graphs, yet it also maintains a history node feature for each node as memory. Specifically, it adopts a multi-layer memory mechanism by keeping memory for both memory and embedding stages. In other words, for the graph embedding part, it also adopts a GRU to combine nodes' previous embedding with the current embedding gathered from updated node memories. All the models update and embed memory for one time at each prediction (i.e., one layer aggregation in equation 3 and equation 4). The node memory dimension is set to 172, and the node embedding dimension is set to 100. Following the training steps in (Rossi et al., 2020), we use Adam optimizer with learning rate $\alpha = 0.01$ to train the models 120 epochs.

**Tasks Details.** Models for node classification are trained to predict binary labels on each node. We use the commonly used Area under the ROC Curve (ROC-AUC) to measure the model performances. The models for edge prediction are self-supervise trained, using the edge information in future steps. During the testing, given a source node, they predict the possibility of whether another node will be its next incoming destination node and then decide which node will be its next neighbor. We use prediction accuracy for evaluating the edge prediction result.

**Dataset Details.** Reddit and Wikipedia are dynamic interaction graphs retrieved from online resources in (Rossi et al., 2020). In Wikipedia datasets, the nodes represent users and wiki pages, and the edges indicate editing from users to pages. In the Reddit dataset, the nodes represent users and subreddits, and an edge within it represents a poster from a user posted on a subreddit. The edge features are represented by text features, and the node labels indicate whether a user is banned. All the abovementioned information is accompanied by timestamps. Align with their original designs (Kumar et al., 2019), and we set the newly input nodes' features as zero feature vectors. Reddit-body and Reddit-title are two larger-scale datasets that represent the directed connections between two subreddits (a subreddit is a community on Reddit). The dataset is collected by SNAP using publicly available Reddit data of 2.5 years from Jan 2014 to April 2017 (Kumar et al., 2018). The statistics of the dataset used are shown in Table 2.

Table 2: Dataset details

| | # of Nodes | # of Edges | # Edge Feature | # of Node Feature |
|---|---|---|---|---|
| **Wikipedia(`WIKI`)** | 9,227 | 157,474 | 172 | 172 |
| **Reddit(`REDDIT`)** | 11,000 | 672,447 | 172 | 172 |
| **Reddit-Body(`REDDIT-BODY`)** | 35,776 | 286,561 | 64 | 172 |
| **Reddit-Title(`REDDIT-TITLE`)** | 54,075 | 571,927 | 64 | 172 |

## B.2 BASELINE ATTACK AND DEFENSES

We adopt the following attacks toward static GNNs. Specifically, we adopt the attack at the same time as our attack time by attacking the existing dynamic graph as a static graph:

**TDGIA (Zou et al., 2021)** is a cutting-edge Graph Injection Attack tailored to compromise static GNNs. This method exploits the inherent vulnerabilities of GNNs and the unique topological characteristics of graphs. In our implementation for each target node, we adhere to the established methodology of TDGIA to identify the top 65 susceptible edges, utilizing their specialized scheme for selecting topologically defective edges. These edges are then optimized using gradient descent. Notably, the scale of modifications applied to each target node in the TDGIA method is substantially larger than our approach, involving adjustments to 65 edges instead of just 3. Furthermore, these modifications will be kept after the attack instead of being removed as our attack.

**Meta_Attack_Heuristic (Li et al., 2022)** is a heuristic-based attack inspired by the meta attack (Zügner & Günnemann, 2019). This heuristic-based approach is an evolution of the original meta-attack, which relied on gradient-based edge selection. The updated heuristic version demonstrates greater versatility across a variety of GNN models and large-scale graphs, and it exhibits enhanced effectiveness compared to its predecessor. Notably, the meta-attack and its heuristic counterpart operate under the assumption that edges lack attributes. Consequently, in our application, we assign an all-zero feature to the fake edges inserted as part of the attack process.

We adopt the following defensive strategies for the vanilla TGNN models:

**Adversarial Training:** In line with the approach detailed in (Madry et al., 2017), we introduce perturbations to the node memories in TGNN models during the training. We then employ a minimax adversarial training scheme to enhance the robustness of the TGNN model against these perturbations.

**Regularization under empirical Lipschitz bound:** Following the methodology in (Jia et al., 2023), we minimize the empirical Lipschitz bound during the TGNN training process, where the empirical Lipschitz bound, $L$, is computed by:

$$L = \sup_{\Delta} \frac{||f(x + \Delta) - f(x)||_2^2}{||\Delta||_2^2} \tag{19}$$

This regularization aims to bound the effectiveness of small perturbations, such as adversarial examples.

Notably, most robust GCN models, such as RobustGCN, SGCN, GraphSAGE, and TAGCN mentioned in (Zou et al., 2021), are primarily tailored for static graph benchmarks. Given their design constraints, these models are unsuited for TGNN setup with dynamic graph benchmarks and do not offer a viable defense for the TGNN models targeted by our attack.

### B.3 EXTRA MAIN RESULTS

Here we report edge prediction accuracies on `REDDIT-TITLE` in Table 3, and node classification AUCs on `WIKI` in Table 4. The results indicate that: (1) The static attacks cannot last long and affect future nods. (2) Our approach can be more and more effective after the attack time

Table 3: Accumulated accuracy of edge prediction in the vanilla/attacked TGNNs over different timestamps on `REDDIT-TITLE`; lower matrices indicate more effective attacks.

| Dataset | | REDDIT-TITLE | | | |
|---|---|---|---|---|---|
| Model | | TGN | Jodie | Dyrep | ROLAND |
| Vanilla | | 0.93 | 0.92 | 0.91 | 0.91 |
| $t_0$ | FN | 0.76 | 0.82 | 0.77 | 0.79 |
| | Meta_h | 0.86 | 0.83 | 0.88 | 0.85 |
| | TDGIA | **0.72** | **0.81** | **0.74** | **0.76** |
| | ours | 0.84 | 0.85 | 0.81 | 0.78 |
| $t_{25}$ | FN | 0.9 | 0.86 | 0.89 | 0.88 |
| | Meta_h | 0.89 | 0.86 | 0.9 | 0.87 |
| | TDGIA | 0.89 | 0.85 | 0.89 | 0.88 |
| | ours | **0.81** | **0.84** | **0.76** | **0.80** |
| $t_{50}$ | FN | 0.9 | 0.86 | 0.9 | 0.88 |
| | Meta_h | 0.9 | 0.86 | 0.9 | 0.88 |
| | TDGIA | 0.89 | 0.86 | 0.9 | 0.87 |
| | ours | **0.77** | **0.82** | **0.76** | **0.77** |

To more comprehensively show the impact of the attack budget, we include detailed results of baselines' and our attacks' effectiveness under the attack budget as 1%. As shown in Table 5, Table 6, and Table 7, our approach can outperform baselines as well, despite fewer nodes being attacked.

Table 4: The AUC of vanilla/attacked TGNNs on the node classification task; lower matrices indicate more effective attacks.

| Dataset | | WIKI | | | |
|---|---|---|---|---|---|
| | Model | TGN | Jodie | Dyrep | ROLAND |
| | Vanilla | 0.90 | 0.88 | 0.89 | 0.90 |
| t0 | FN | **0.77** | 0.87 | **0.75** | 0.78 |
| | Meta | 0.86 | 0.83 | 0.86 | 0.85 |
| | TDGIA | 0.73 | **0.82** | 0.76 | **0.75** |
| | ours | 0.82 | 0.88 | 0.84 | 0.80 |
| t25 | FN | 0.90 | 0.88 | 0.88 | 0.88 |
| | Meta | 0.89 | 0.87 | 0.88 | 0.89 |
| | TDGIA | 0.88 | 0.87 | 0.88 | 0.89 |
| | ours | **0.82** | **0.85** | **0.81** | **0.79** |
| t50 | FN | 0.90 | 0.88 | 0.88 | 0.90 |
| | Meta | 0.90 | 0.89 | 0.90 | 0.90 |
| | TDGIA | 0.90 | 0.88 | 0.88 | 0.90 |
| | ours | **0.80** | **0.85** | **0.77** | **0.77** |

Table 5: Accumulated accuracy of edge prediction in the vanilla/attacked TGNNs over different timestamps on WIKI and REDDIT; The attack budget is 1% for all attacks; lower matrices indicate more effective attacks.

| Dataset | | WIKI | | | | REDDIT | | | |
|---|---|---|---|---|---|---|---|---|---|
| | Model | TGN | Jodie | Dyrep | ROLAND | TGN | Jodie | Dyrep | ROLAND |
| | Vanilla | 0.93 | 0.87 | 0.85 | 0.94 | 0.96 | 0.98 | 0.96 | 0.95 |
| $t_0$ | FN | 0.89 | 0.83 | 0.82 | 0.85 | 0.93 | 0.93 | 0.92 | 0.85 |
| | Meta | 0.92 | 0.85 | 0.83 | 0.89 | 0.95 | 0.96 | 0.94 | 0.93 |
| | TDGIA | 0.83 | 0.81 | 0.77 | 0.83 | 0.89 | 0.88 | 0.88 | 0.8 |
| | ours | 0.9 | 0.85 | 0.86 | 0.9 | 0.93 | 0.94 | 0.94 | 0.86 |
| $t_{25}$ | FN | 0.92 | 0.87 | 0.85 | 0.94 | 0.97 | 0.97 | 0.96 | 0.95 |
| | Meta | 0.93 | 0.86 | 0.85 | 0.93 | 0.95 | 0.98 | 0.95 | 0.94 |
| | TDGIA | 0.91 | 0.84 | 0.83 | 0.93 | 0.94 | 0.96 | 0.96 | 0.92 |
| | ours | 0.8 | 0.8 | 0.78 | 0.88 | 0.81 | 0.84 | 0.91 | 0.84 |
| $t_{50}$ | FN | 0.94 | 0.87 | 0.86 | 0.94 | 0.97 | 0.97 | 0.96 | 0.95 |
| | Meta | 0.94 | 0.87 | 0.86 | 0.93 | 0.96 | 0.98 | 0.95 | 0.95 |
| | TDGIA | 0.94 | 0.87 | 0.85 | 0.93 | 0.96 | 0.97 | 0.95 | 0.93 |
| | ours | 0.85 | 0.83 | 0.81 | 0.86 | 0.83 | 0.84 | 0.91 | 0.83 |

Table 6: Accumulated accuracy of edge prediction in the vanilla/attacked TGNNs over different timestamps on REDDIT-BODY and REDDIT-TITLE; The attack budget is 1% for all attacks; lower matrices indicate more effective attacks.

| Dataset | | REDDIT-BODY | | | | REDDIT-TITLE | | | |
|---|---|---|---|---|---|---|---|---|---|
| | Model | TGN | Jodie | Dyrep | ROLAND | TGN | Jodie | Dyrep | ROLAND |
| | Vanilla | 0.9 | 0.87 | 0.9 | 0.88 | 0.93 | 0.92 | 0.91 | 0.91 |
| $t_0$ | FN | 0.85 | 0.85 | 0.81 | 0.83 | 0.88 | 0.88 | 0.85 | 0.83 |
| | Meta | 0.87 | 0.85 | 0.87 | 0.86 | 0.92 | 0.89 | 0.89 | 0.9 |
| | TDGIA | 0.81 | 0.83 | 0.79 | 0.78 | 0.85 | 0.87 | 0.85 | 0.83 |
| | ours | 0.87 | 0.85 | 0.85 | 0.82 | 0.88 | 0.9 | 0.86 | 0.85 |
| $t_{25}$ | FN | 0.9 | 0.84 | 0.89 | 0.88 | 0.92 | 0.92 | 0.9 | 0.91 |
| | Meta | 0.9 | 0.87 | 0.9 | 0.88 | 0.93 | 0.93 | 0.91 | 0.91 |
| | TDGIA | 0.88 | 0.86 | 0.9 | 0.87 | 0.92 | 0.92 | 0.9 | 0.91 |
| | ours | 0.84 | 0.86 | 0.8 | 0.82 | 0.85 | 0.88 | 0.81 | 0.86 |
| $t_{50}$ | FN | 0.9 | 0.87 | 0.9 | 0.88 | 0.93 | 0.92 | 0.9 | 0.91 |
| | Meta | 0.9 | 0.88 | 0.9 | 0.88 | 0.93 | 0.93 | 0.9 | 0.91 |
| | TDGIA | 0.89 | 0.87 | 0.9 | 0.87 | 0.93 | 0.91 | 0.9 | 0.9 |
| | ours | 0.79 | 0.85 | 0.77 | 0.83 | 0.8 | 0.83 | 0.82 | 0.83 |

Table 7: The AUC of vanilla/attacked TGNNs on the node classification task under 1% node attacked budget; lower matrices indicate more effective attacks.

| Dataset | | **WIKI** | | | |
|---------|-------|-------|-------|-------|--------|
| | **Model** | **TGN** | **Jodie** | **Dyrep** | **ROLAND** |
| | Vanilla | 0.9 | 0.88 | 0.89 | 0.9 |
| $t_0$ | FN | 0.83 | 0.88 | 0.83 | 0.83 |
| | Meta | 0.87 | 0.85 | 0.88 | 0.88 |
| | TDGIA | 0.81 | 0.85 | 0.83 | 0.8 |
| | ours | 0.86 | 0.88 | 0.86 | 0.85 |
| $t_{25}$ | FN | 0.89 | 0.88 | 0.89 | 0.9 |
| | Meta | 0.9 | 0.88 | 0.88 | 0.89 |
| | TDGIA | 0.9 | 0.87 | 0.89 | 0.89 |
| | ours | 0.82 | 0.85 | 0.81 | 0.81 |
| $t_{50}$ | FN | 0.9 | 0.87 | 0.89 | 0.9 |
| | Meta | 0.9 | 0.88 | 0.89 | 0.9 |
| | TDGIA | 0.9 | 0.87 | 0.89 | 0.89 |
| | ours | 0.82 | 0.88 | 0.79 | 0.82 |

### B.4 EXTRA RESULTS ON ATTACKS UNDER DEFENSES

We include the results of two attacks, i.e., FakeNode and MemStranding, under the two defenses, i.e., `adv_train` and `Lip_reg`, on two TGNN models, i.e., Jodie and Dyrep. The observations are similar to the prior analysis.

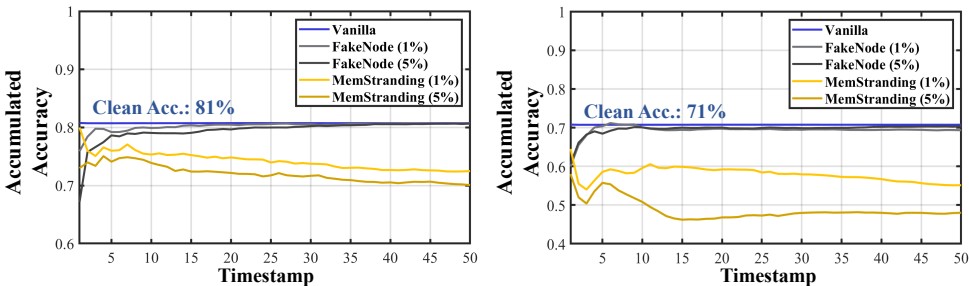

Figure 12: Accumulated accuracies of DyRep under `Adv_train`(left), and `Lip_reg`(right) with FakeNode and our attack on `WIKI`.

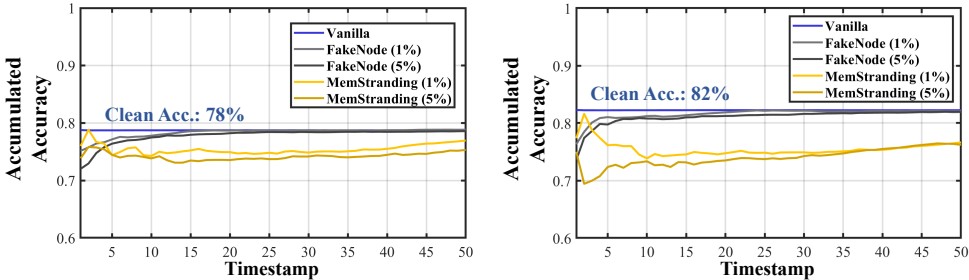

Figure 13: Accumulated accuracies of JODIE under `Adv_train`(left), and `Lip_reg`(right) with FakeNode and our attack on `WIKI`.

## B.5 EXTRA ABLATION STUDY

We include the results for the ablation studies under the TGN model and `REDDIT` dataset in Figure B.5. The results show a similar pattern as we observed in Section 5.

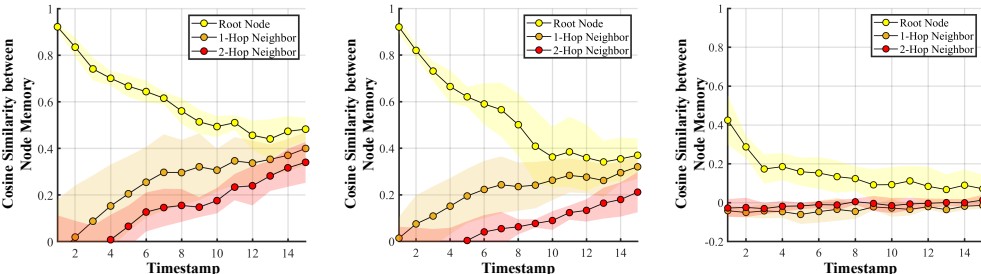

Figure 14: The similarities between root nodes' initial noisy memories (at the time of the attack) and themselves'/their subsequent neighbors' memories in MemSranding w/o (left) converge state, MemSranding w/o persisting loss (middle), and regular nodes (right). All results are from the TGN model and `REDDIT` dataset.

## B.6 EXTRA SENSITIVITY STUDY

We include more results for different target node sampling strategies and attack budgets in Figure B.6. The results show a similar pattern as we observed in Section 5.

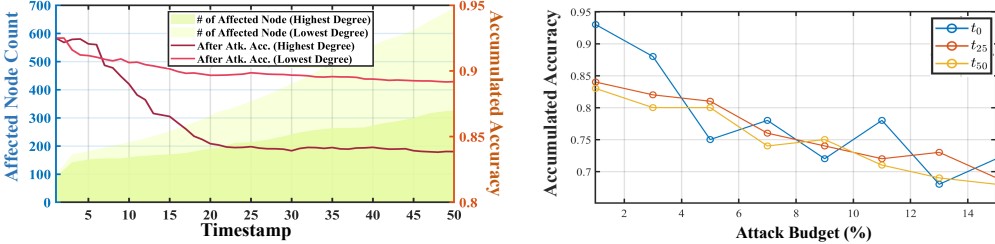

Figure 15: (left) Comparison between two strategies for selecting the injected node: lowest degree and highest degree nodes. Count of affected nodes and overall accuracy over time. (RIGHT) The accumulated accuracy at $t_0$, $t_{25}$, and $t_{50}$ under different attack budgets (% of total nodes). All results above are from TGN and `REDDIT`

## B.7   ACCUMULATED ACCURACIES OVER TIME ON DIVERSE MODELS

We report the accumulated accuracies over time collected from Jodie and Dyrep on the `WIKI` and `REDDIT` datasets. The results include model accuracies under the vanilla (i.e., un-attacked), baseline (i.e., FakeNode), and our (i.e., MemStranding) attacks in edge prediction tasks.

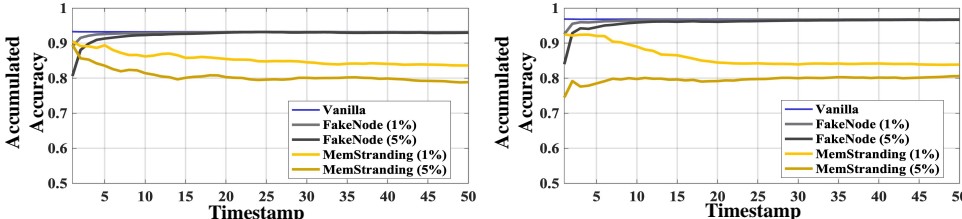

Figure 16: Accumulated accuracies of TGN under different attacks in link prediction tasks over time in `WIKI` (left) and `REDDIT` (right) datasets.

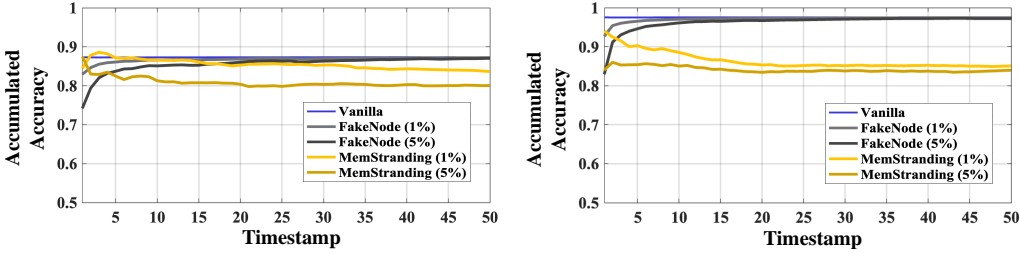

Figure 17: Accumulated accuracies of Jodie under different attacks in link predictions over time with `WIKI` (left) and `REDDIT` (right) datasets.

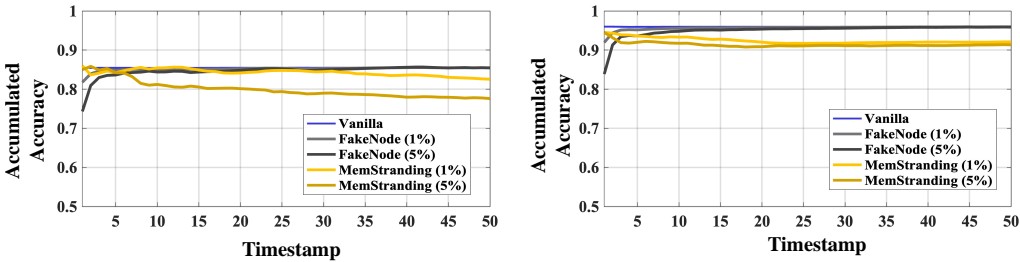

Figure 18: Accumulated accuracies of Dyrep under different attacks in link predictions over time with `WIKI` (left) and `REDDIT` (right) datasets.

## B.8 AFFECTED NODES

We report the number and accumulated accuracies over time of affected nodes over time in Jodie and Dyrep on the WIKI and REDDIT datasets. The results include model accuracies under our (i.e., MemStranding) attack in edge prediction tasks.

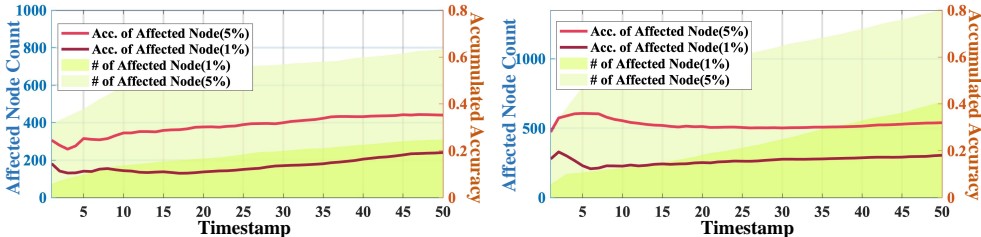

Figure 19: Count of affected nodes (presented as the colored areas) and their accumulated accuracies (presented as lines) in WIKI (left) and REDDIT (right) over time. The data are collected in TGN.

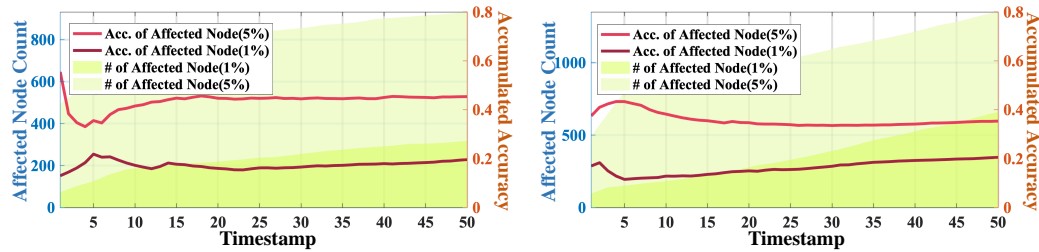

Figure 20: Count of affected nodes (presented as the colored areas) and their accumulated accuracies (presented as lines) in WIKI (left) and REDDIT (right) over time. The data are collected in Jodie.

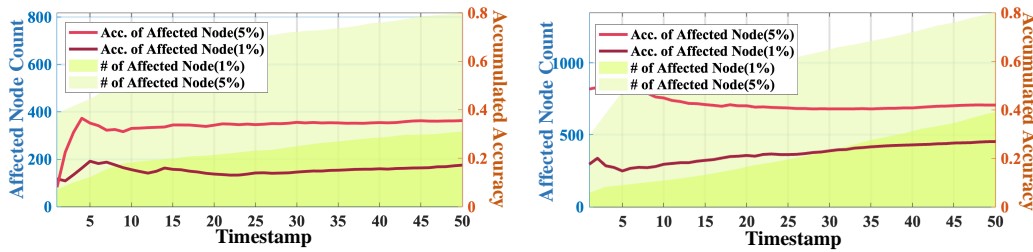

Figure 21: Count of affected nodes (presented as the colored areas) and their accumulated accuracies (presented as lines) in WIKI (left) and REDDIT (right) over time. The data are collected in Dyrep.

## B.9 NOISE PROPAGATING

We report the cosine similarities between the initial root node and its neighbors over time in Jodie and Dyrep on the `WIKI` and `REDDIT` datasets. The results include similarities under our (i.e., MemStranding) attack in edge prediction tasks.

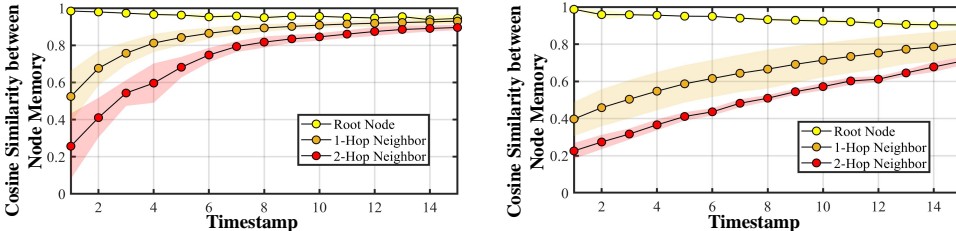

Figure 22: The similarities between root nodes' initial noisy memories (at the time of the attack) and themselves'/their subsequent neighbors' memories in `WIKI` (left) and `REDDIT` (right) over time. The data are collected in TGN.

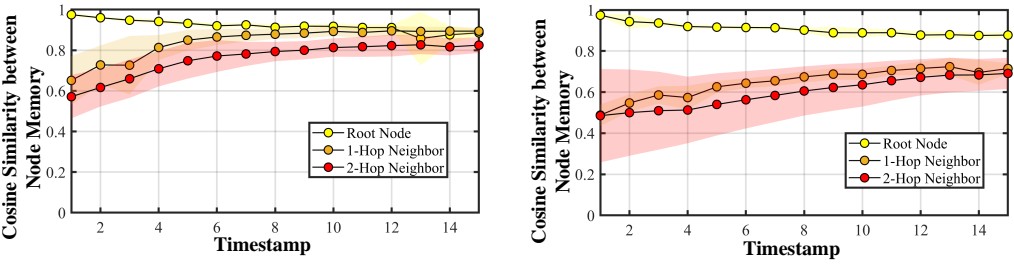

Figure 23: The cosine similarities between root nodes' initial memory (at the time of the attack) and themselves/their subsequent neighbors' memories in `WIKI` (left) and `REDDIT` (right) over time. The data are collected in Jodie.

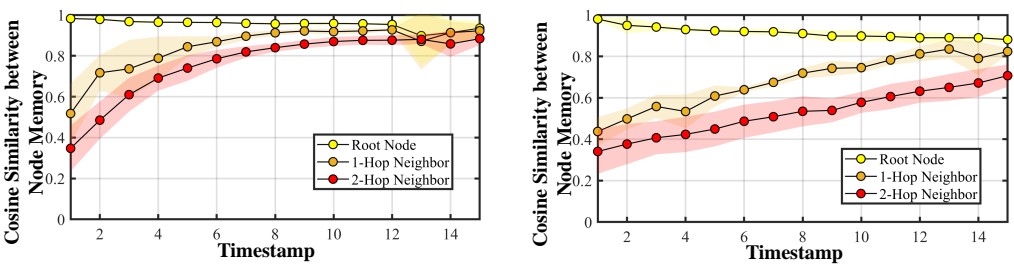

Figure 24: The cosine similarities between root nodes' initial memory (at the time of the attack) and themselves/their subsequent neighbors' memories in `WIKI` (left) and `REDDIT` (right) over time. The data are collected in Dyrep.

## C Discussion And Future Work

### C.1 Limits under different models and graphs.

While the experiment results in Appendix B.3 and Appendix B.6 demonstrated that MemStranding can be well-generalized on various inputs, several limitations can be observed according to the performance variance between different models. While our approach can effectively mislead TGN, ROLAND, and DyRep, its effectiveness is less significant on JODIE, which uses differences between a node's current and its last update time to decay the memory. From these observations, we deduce that our attack may encounter limitations in two specific scenarios:

- *Limited Influence of Node Memory on Predictions:* Our attack's effectiveness may be mitigated in situations where the node memory has a relatively minor role in influencing the model's predictions.
- *Usage of Additional Information in TGNN Models*: The effectiveness may also be constrained when the targeted TGNN model incorporates additional information beyond the node memory for its predictive processes.

While our attack strategy outperforms the baselines, these insights highlight potential limitations under certain model-specific conditions.

### C.2 Potential Defenses.

While we demonstrate that many existing defense schemes, such as adversarial training or regularization, are less effective on our attacks, we expect a potential attack-oriented defense scheme for our attack using memory filtering. Specifically, a potential defensive approach for our attack is to pay less attention to the nodes' memory and rely more on their current input adaptively.

This scheme stems from the observation that our attacks are less effective on JODIE in node classification tasks. One key difference in JODIE is that it decays the node memory based on the time differences between the prediction time and the node's last update time. This mechanism introduces more hints (i.e., time differences) in addition to the memory itself, which cannot be effectively distorted by the attacks and yields some crucial information. For example, a Wikipedia user is less likely to be banned if he/she makes a new post after being inactive for a long while.

Therefore, using this non-memory information or current information that does not interact with node memory could effectively hinder adversarial noises. To this end, an intelligent defense mechanism can judiciously filter out the memory and adaptively focus more on non-memory information if the memory is suspicious or potentially noisy.

