# OpenReview forum: "MemStranding: Adversarial attacks on temporal graph neural networks"
_ICLR.cc/2024/Conference — Submitted to ICLR 2024_

### Official Review · Reviewer_4nwA · 2023-10-31

**Soundness:** 3 good
**Presentation:** 3 good
**Contribution:** 3 good
**Rating:** 6
**Confidence:** 4

**Summary:**

This paper focuses on the problem of adversarial attack on temporal graph neural networks.  Two challenges (e.g., Noise Decaying and Knowledge Missing) in attacking  temporal GNNs are first proposed. To address these challenges, the authors proposes an effective adversarial attack framework called MemStranding. Experimental results show that MemStranding can significantly decrease the TGNN models’ performances in various tasks.

**Strengths:**

1. The paper focuses on adversarial attacks on TGNNs, studying the robustness of TGNNs to benefit their applications.
2. The authors provide a clear explanation of why typical attacks fail in dynamic graph scenarios.
3. In response to the failure of typical attacks, two attack goals are proposed: noise persisting and noise propagating, effectively addressing the issue.

**Weaknesses:**

1. The studied problem is interesting but the application scenarios of attacking TGNNs should be introduced.
2. The paper provides a good explanation of why typical attacks fail, but does not provide experimental results. I am curious about the effectiveness of typical attacks on dynamic graphs.
3. Section 4.3, Stage 1, lacks further analysis on victim node sampling. The choice of high-degree nodes as root nodes is explained, but there is no analysis of what would happen if low-degree nodes were chosen as root nodes. Moreover, are there criteria for selecting support nodes?

**Questions:**

1. If the persisting loss is removed, how would the similarity between the root node's memory and the initial noisy memory change in Figure 8? I'm curious about this.
2. Still regarding Figure 8, how does the similarity of memory between the root node and its 1-hop and 2-hop neighbors change during normal training of TGNNs?
3. The noise persisting loss and noise propagating loss proposed in this paper effectively prolong the efficacy of noise, even after multiple new actions. I'm curious about the effect if we combine the attack loss proposed in this paper with meta attack.
4. Question about Section 4.3, Stage 2. The paper mentions, 'Lastly, we can add the solved noisy message as a fake node or fake edge accordingly and remove it after the attack.'  Is this similar to directly modifying the victim node's memory? Will this noisy message's impact spread with the arrival of the next action?

[1] Adversarial Attacks on Graph Neural Networks via Meta Learning, ICLR 2019.

---

> ### Author Response · Authors · 2023-11-20
> **Author Response to Reviewer 4nWA (1/3)**
>
> ## **Comments**
> We would like to thank the reviewer for the positive feedback and valuable suggestions. We appreciate the reviewer's acknowledgment that our proposed work is interesting and clearly presented. We carefully addressed all the reviewers' questions and revised the paper accordingly. We hope our response can help clarify the reviewer's questions.
>
> ## **Q1. Application scenarios of attacking TGNNs**
>
> Thanks for the comment and valuable suggestion. The application scenarios of attacking TGNNs could exist in many dynamic graph prediction tasks. For instance, for social media such as Reddit, there could be node classification tasks that predict if a user/subreddit will be banned or not using TGNNs. The adversarial attacks toward these models can cause an innocent user/subreddit to always be banned mistakenly or prevent a malicious user/subreddit from being banned even if it always gives junk information, which causes severe benefits for attackers. We also add a brief example of this application scenario in **Section 1** of our revised paper.
>
> ## **Q2. Effectiveness of typical static attacks on dynamic graphs:**
>
> Thanks for the question. We would like to mention our baseline (i.e., FakeNode[1]) is a typical static GNN attack. As highlighted in **Section 2, Table 1, and Figure 6** in the original paper, while this static GNN attack can affect its target nodes and lead to significant performance degradation, the noise cannot persist in the target nodes and propagate to more victims.
>
>
> We also include more static attacks, including **TDGIA**[2] and **Meta-Attack-Heuristic**[3], for a more comprehensive evaluation. The results are shown in **Table R.10**. Typical static attacks can effectively mislead TGNN models at the attack time; however, the effectiveness drops drastically after a few updates. These results indicate that typical static attacks can hardly work for dynamic graphs due to future updates on the victim nodes. We also added the results to **Table 1 in Section 5.2 and Tables 3&5 in Appendix B.3** in our revised paper.
>
> >**Table R.10.:** Accumulated accuracy over different timestamps of the vanilla/attacked TGNNs in WIKI and REDDIT.
> | Dataset |       | Wiki |       |       |        |   | Reddit |       |       |        |
> |---------|-------|------|-------|-------|--------|---|--------|-------|-------|--------|
> | **Model**   |       | **TGN** | **Jodie** | **Dyrep** | **Roland** |   | **TGN** | **Jodie** | **Dyrep** | **Roland** |
> | Vanilla Acc. |       | 0.93 | 0.87  | 0.85  | 0.94   |   | 0.96   | 0.98  | 0.96  | 0.95   |
> | $t_0$   | FN    | 0.81 | 0.74  | 0.74  | 0.82   |   | 0.84   | 0.83  | 0.84  | 0.83   |
> |         | Meta  | 0.9  | 0.83  | 0.81  | 0.85   |   | 0.93   | 0.95  | 0.9   | 0.92   |
> |         | TDGIA | 0.77 | 0.72  | 0.71  | 0.8    |   | 0.74   | 0.8   | 0.81  | 0.74   |
> |         | ours  | 0.89 | 0.88  | 0.85  | 0.87   |   | 0.75   | 0.84  | 0.94  | 0.82   |
> | $t_{25}$  | FN    | 0.92 | 0.87  | 0.85  | 0.94   |   | 0.97   | 0.97  | 0.96  | 0.93   |
> |         | Meta  | 0.93 | 0.87  | 0.84  | 0.93   |   | 0.96   | 0.98  | 0.94  | 0.96   |
> |         | TDGIA | 0.93 | 0.81  | 0.84  | 0.92   |   | 0.94   | 0.95  | 0.95  | 0.9    |
> |         | ours  | 0.8  | 0.8   | 0.78  | 0.85   |   | 0.81   | 0.84  | 0.91  | 0.8    |
> | $t_{50}$   | FN    | 0.94 | 0.87  | 0.86  | 0.94   |   | 0.97   | 0.97  | 0.96  | 0.95   |
> |         | Meta  | 0.93 | 0.87  | 0.85  | 0.93   |   | 0.97   | 0.98  | 0.94  | 0.95   |
> |         | TDGIA | 0.93 | 0.87  | 0.85  | 0.93   |   | 0.96   | 0.97  | 0.95  | 0.92   |
> |         | ours  | 0.75 | 0.81  | 0.76  | 0.84   |   | 0.8    | 0.84  | 0.91  | 0.8    |
>
>
> [1] Wang, Xiaoyun, et al. "Fake node attacks on graph convolutional networks." Journal of Computational and Cognitive Engineering 1.4 (2022): 165-173.
>
> [2]  Zou, Xu, et al. "Tdgia: Effective injection attacks on graph neural networks." Proceedings of the 27th ACM SIGKDD Conference on Knowledge Discovery & Data Mining. 2021.
>
> [3] Li, Kuan, et al. "Revisiting graph adversarial attack and defense from a data distribution perspective." The Eleventh International Conference on Learning Representations. 2022.

---

> ### Author Response · Authors · 2023-11-20
> **Author Response to Reviewer 4nWA (2/3)**
>
> ## **Q3. Analysis of target/support node sampling strategies**
>
> Thanks for the suggestion. To further analyze the impact of victim node sampling, we compare the numbers of affected nodes and the resulting accuracy of selecting target nodes from the highest and lowest degree. The results are shown in **Table R.11**. As one can observe, the effectiveness of our attack is weakened in these cases. The main reason is that these low-degree nodes can only propagate noises to fewer nodes across the graph. Specifically, the affected nodes are significantly lower than the highest-degree first strategy. The main reason is that these low-degree nodes can only propagate noises to fewer nodes across the graph.
>
> > **Table R.11.** Number of affected nodes and accumulated accuracies of our attack under selecting 1% highest/lowest-degree victim nodes.
> |                |                    | t0   | t10  | t20  | t30  | t40  | t50  |
> |----------------|--------------------|------|------|------|------|------|------|
> | **Highest Degree** | # of Affected Node | 74   | 176  | 211  | 253  | 285  | 313  |
> |                | Overall Acc.       | 0.91 | 0.88 | 0.86 | 0.86 | 0.85 | 0.85 |
> |                |                    |      |      |      |      |      |      |
> | **Lowest Degree**  | # of Affected Node | 79   | 134  | 159  | 174  | 192  | 211  |
> |                | Overall Acc.       | 0.83 | 0.88 | 0.87 | 0.87 | 0.87 | 0.86 |
>
>
> However, it is noteworthy that even when targeting low-degree nodes, our attack still induces a significant drop in accuracy. This effect can be attributed to the fact that the injected noise, originating from low-degree nodes, propagates to their connected high-degree neighbors, maintaining sufficient potency to impact the network.
>
>
> For the support node sampling, we also sample their highest-degree neighbors for the same purpose. We clarify this in **Section 4.3** in our revised paper.
>
>
>
> ## **Q4~Q5. Similarities w/o persisting loss and in vanilla TGNNs.**
>
> Thanks for the question. To conduct a more comprehensive study, we compare the cosine similarity between the memories of the root nodes at $t_0$ with those in themselves and their one-hop and two-hop neighbors at each timestamp after the attack in three versions: (1) MemStranding, (2 )MemStranding w/o persisting loss, and (3) those similarities in the original TGNN without attacks. As shown in **Table R.12**, the similarity between nodes' initial attacked memory and their future counterparts drops drastically in the normal TGNNs and fails to propagate to their neighbors and persist them. If the persisting loss is not enabled, the similarities also suffer drops and achieve much lower similarities in the future. We also include more results in **Figure 7 in Section 5.2, and Figure 14 in Appendix B.5** in our revised paper.
>
> >**Table R.12.** Similarities between nodes in (1) MemStranding, (2 )MemStranding w/o persisting loss, and (3) those similarities in the original TGNN without attacks.
> |                   |                  | t1    | t3   | t5   | t7   | t9   | t11  | t13  | t15  |
> |-------------------|------------------|-------|------|------|------|------|------|------|------|
> | **MemStranding** | Root Node        | 0.98  | 0.97 | 0.96 | 0.96 | 0.96 | 0.95 | 0.95 | 0.94 |
> |                   | One-Hop Neighber | 0.52  | 0.76 | 0.84 | 0.88 | 0.90 | 0.91 | 0.92 | 0.93 |
> |                   | Two-Hop Neighber | 0.26  | 0.54 | 0.68 | 0.79 | 0.83 | 0.86 | 0.89 | 0.90 |
> |                   |                  |       |      |      |      |      |      |      |      |
> | **MemStranding W/O Persist Loss**  | Root Node        | 0.86  | 0.77 | 0.74 | 0.70 | 0.69 | 0.68 | 0.69 | 0.67 |
> |                   | One-Hop Neighber | 0.02  | 0.26 | 0.39 | 0.52 | 0.57 | 0.60 | 0.61 | 0.63 |
> |                   | Two-Hop Neighber | -0.02 | 0.00 | 0.08 | 0.29 | 0.43 | 0.48 | 0.52 | 0.57 |
> |                   |                  |       |      |      |      |      |      |      |      |
> | **Normal TGNN**       | Root Node        | 0.39  | 0.12 | 0.06 | 0.05 | 0.01 | 0.01 | 0.01 | 0.00 |
> |                   | One-Hop Neighber | 0.00  | 0.03 | 0.04 | 0.08 | 0.10 | 0.08 | 0.09 | 0.09 |
> |                   | Two-Hop Neighber | 0.01  | 0.08 | 0.01 | 0.01 | 0.02 | 0.01 | 0.02 | 0.02 |

---

> ### Author Response · Authors · 2023-11-20
> **Author Response to Reviewer 4nWA (3/3)**
>
> ## **Q6. The potential of combining our approach with the Meta Attack**
>
> Thanks for the insightful question. It could be exciting to combine our approach with the meta-attack [4]. However, combining meta-attack with our approach is non-trivial for two reasons: (1) it is tailored for the GCN model; however, the approach is hard to migrate to TGNN, which adopts different message aggregation schemes. (2) it adds/deletes edges with no edge features; however, in our cases, the edge contains many features (e.g., 300 dimensions).
>
> Therefore, we leverage insights from an enhanced meta-attack version mentioned in [5] to evaluate the potential of combining our approach with the meta-attack. Specifically, we **(1) adopt its meta-attack-based heuristic to select the most effective fake edges (i.e., victim nodes and fake messages) and then (2) utilize our loss to solve the fake messages. The results are shown in Table R.13.**
>
> > **Table R.13.** Results of Meta-h, MemStranding (Ours), and combined attack (Meta-h + Ours)
> | **Dataset** |             | **WIKI** |       |       |    | **REDDIT** |       |       |
> |---------|-------------|------|-------|-------|----|--------|-------|-------|
> | **Model**   |             | **TGN**  | **Jodie** | **Dyrep** | 　 | **TGN**  | **Jodie** | **Dyrep** |
> | Vanilla |             | 0.93 | 0.87  | 0.85  | 　 | 0.96   | 0.98  | 0.96  |
> | $t_0$      | Meta-h        | 0.92 | 0.85  | 0.83  | 　 | 0.95   | 0.96  | 0.94  |
> |         | ours        | 0.90 | 0.85  | 0.86  | 　 | 0.93   | 0.94  | 0.94  |
> |         | Meta-h + Ours | 0.86 | 0.81  | 0.80  |    | 0.90   | 0.90  | 0.91  |
> |         |             |      |       |       |    |        |       |       |
> | $t_{25}$     | Meta-h        | 0.93 | 0.87  | 0.84  | 　 | 0.96   | 0.98  | 0.94  |
> |         | ours        | 0.80 | 0.80  | 0.78  | 　 | 0.81   | 0.84  | 0.91  |
> |         | Meta-h + Ours | 0.79 | 0.81  | 0.78  |    | 0.78   | 0.82  | 0.90  |
> |         |             |      |       |       |    |        |       |       |
> | $t_{50}$     | Meta-h        | 0.93 | 0.87  | 0.85  | 　 | 0.97   | 0.98  | 0.94  |
> |         | ours        | 0.75 | 0.81  | 0.76  | 　 | 0.80   | 0.84  | 0.91  |
> |         | Meta-h + Ours | 0.76 | 0.80  | 0.79  |    | 0.80   | 0.82  | 0.90  |
>
>
> As we can observe, the combined approach shows better results. The reasons are twofold: First, the added message is not removed after the attack; hence, it keeps affecting victim nodes. Second, the Meta-h attack uses a more advanced scheme to find victim nodes instead of our highest-degree policy.
>
> [4] Zügner, D., and S. Günnemann. "Adversarial attacks on graph neural networks via meta learning.< i> ICLR</i>." (2019).
>
> [5] Li, Kuan, et al. "Revisiting graph adversarial attack and defense from a data distribution perspective." The Eleventh International Conference on Learning Representations. 2022.
>
>
> ## **Q7. The impact of noisy messages after future actions.**
>
> Thanks for the question, and sorry for the unclear explanation. For the first question, solving and inserting noisy messages is similar to modifying the victim node’s memory. Since it is specifically solved to forge the victim’s node memory into the expected format, we opted for this injection method due to practical constraints: direct modification of a node’s memory might not be feasible, particularly if the memory is stored server-side and requires specific authorization to alter. **Conversely, generating fake messages through fake comments or users is more feasible and realistic.**
>
> **The noisy message’s impact will not spread with the arrival of the next action since we will remove it after the attack, as we highlighted in Figure 5** in the original paper. In such cases, the fake message itself will no longer be visible and participate in the future updates of this node. For example, we may delete the fake comments or have the fake user unfollow a post after a few days. However, the impact of the noisy message will be injected into the victim node’s memory and then spread to all future updates involving this node’s memory.

---

> ### Comment · Reviewer_4nwA · 2023-11-21
> **Response to Authors**
>
> Dear Authors，
>
> Thanks for the detailed responses. Most of my concerns have been addressed. I keep my rating unchanged.

---

> > ### Author Response · Authors · 2023-11-22
> > **Thanks for the positive feedback**
> >
> > We would like to thank you again for your time and your positive rating. This is a great affirmation of our work.

---

### Official Review · Reviewer_LiVK · 2023-10-31

**Soundness:** 3 good
**Presentation:** 3 good
**Contribution:** 3 good
**Rating:** 6
**Confidence:** 2

**Summary:**

This paper introduces the MemStranding framework, which is designed to launch adversarial attacks on Temporal Graph Neural Networks (TGNNs) by utilizing node memories to generate persistent and propagating adversarial perturbations in dynamic graphs. The authors empirically validate the efficacy of MemStranding in diminishing the performance of TGNN models across a spectrum of tasks.

**Strengths:**

1. This paper stands out for its innovative approach to attacking TGNNs. It identifies the limitations of existing adversarial methods and introduces a novel framework that uses node memories to create persistent and spreading adversarial disturbances in dynamic graphs, an unexplored concept in prior research.

2. The paper is lucidly written and easily comprehensible. It offers clear explanations of the paper's concepts and techniques, ensuring accessibility to a broad readership.

3. This paper makes a significant contribution by introducing a previously unexplored approach to TGNN attacks. The proposed framework is practical and effective, as demonstrated through compelling experimental results in various scenarios. The paper's findings carry vital implications for TGNN model security and advocate for further research in this domain.

**Weaknesses:**

1. The paper lacks a comprehensive discussion of the limitations of the proposed method, including performance variations with different TGNN architectures and graph data characteristics.

2. It is not clear that fakenode is the state-of-the-art attack method. However, the paper only compare the proposed method with fakenode only.

**Questions:**

1. Can you provide more insights into the limitations of the proposed method, such as its sensitivity to the size and density of the graph, the number of target nodes, and the choice of the attack budget?

2. Can you provide more insights into the potential defenses against the proposed method, such as the use of adversarial training, graph regularization, or outlier detection?

---

> ### Author Response · Authors · 2023-11-20
> **Author Response to Reviewer LiVK (1/4)**
>
> ## **Comments**
> We would like to thank the reviewer for the positive feedback and valuable suggestions. We appreciate the reviewer's acknowledgment that our proposed work is novel and significant. We carefully address all the reviewer’s questions and revise the paper accordingly, including more discussion on our attack’s limitations and its potential defenses. Additionally, we implemented and compared more state-of-the-art GNN attacks to our approach. We hope our response can help clarify the reviewer's questions.
>
> ## **Q1. Comparisons with more state-of-the-art adversarial attacks on diverse datasets and TGNN architectures**
>
> Thank you for the kind suggestion. We add more comparisons as following.
>
> - #### **Additional Baseline Attacks and TGNN model:**
>
> To provide more comprehensive comparisons, we compare our work with two more recent state-of-the-art attacks for static GNNs: **TDGIA**[1] and **Meta-Attack-Heuristic (Meta-h)** mentioned in [2]. Additionally, as suggested, we also include the evaluation of our work in conjunction with **ROLAND**[3]. We show the results under the 5% node attack budget in **Table R.5.** The results indicate that our attack can outperform various baselines on more recent TGNN models.
>
> >**Table R.5.:** Accumulated accuracy over different timestamps of the vanilla/attacked TGNNs in WIKI and REDDIT.
> | **Dataset** |       | **WIKI** |       |       |        |   | **REDDIT** |       |       |        |
> |---------|-------|------|-------|-------|--------|---|--------|-------|-------|--------|
> | **Model**   |       | **TGN** | **Jodie** | **Dyrep** | **Roland** |   | **TGN** | **Jodie** | **Dyrep** | **Roland** |
> | Vanilla Acc. |       | 0.93 | 0.87  | 0.85  | 0.94   |   | 0.96   | 0.98  | 0.96  | 0.95   |
> | $t_0$   | FN    | 0.81 | 0.74  | 0.74  | 0.82   |   | 0.84   | 0.83  | 0.84  | 0.83   |
> |         | Meta-h  | 0.9  | 0.83  | 0.81  | 0.85   |   | 0.93   | 0.95  | 0.9   | 0.92   |
> |         | TDGIA | 0.77 | 0.72  | 0.71  | 0.8    |   | 0.74   | 0.8   | 0.81  | 0.74   |
> |         | ours  | 0.89 | 0.88  | 0.85  | 0.87   |   | 0.75   | 0.84  | 0.94  | 0.82   |
> | $t_{25}$  | FN    | 0.92 | 0.87  | 0.85  | 0.94   |   | 0.97   | 0.97  | 0.96  | 0.93   |
> |         | Meta-h  | 0.93 | 0.87  | 0.84  | 0.93   |   | 0.96   | 0.98  | 0.94  | 0.96   |
> |         | TDGIA | 0.93 | 0.81  | 0.84  | 0.92   |   | 0.94   | 0.95  | 0.95  | 0.9    |
> |         | ours  | 0.8  | 0.8   | 0.78  | 0.85   |   | 0.81   | 0.84  | 0.91  | 0.8    |
> | $t_{50}$   | FN    | 0.94 | 0.87  | 0.86  | 0.94   |   | 0.97   | 0.97  | 0.96  | 0.95   |
> |         | Meta-h  | 0.93 | 0.87  | 0.85  | 0.93   |   | 0.97   | 0.98  | 0.94  | 0.95   |
> |         | TDGIA | 0.93 | 0.87  | 0.85  | 0.93   |   | 0.96   | 0.97  | 0.95  | 0.92   |
> |         | ours  | 0.75 | 0.81  | 0.76  | 0.84   |   | 0.8    | 0.84  | 0.91  | 0.8    |
>
> We also include more results in **Table 1 in Section 5.2 and Table 5&7 in Appendix B.3** in our revised paper.

---

> ### Author Response · Authors · 2023-11-20
> **Author Response to Reviewer LiVK (2/4)**
>
> - #### **Additional Dataset:**
>
> To measure if our approach can generalize the larger real-world applications, we evaluate our works on two larger datasets from real-world social media: **Reddit-body** and **Reddit-title** [4].
> Compared to WIKI and REDDIT, which contain approximately 10,000 nodes, these datasets have more nodes (i.e., 35,000 and 54,000 nodes, respectively). The details of the datasets are introduced in **Table 2 in Appendix B.1** in our revised paper.
>
>
> We show the results under the 5% node attack budget in **Table R.6**. The results indicate that our attack can outperform baselines on larger graphs.
>
>
> >**Table R.6** Accumulated accuracy over different timestamps of the vanilla/attacked TGNNs in Reddit-body and Reddit-title.
> | **Dataset** |       | **Reddit-body** |       |       |        |   | **Reddit-title** |       |       |        |
> |---------|-------|-------------|-------|-------|--------|---|--------------|-------|-------|--------|
> | **Model**   |       | **TGN** | **Jodie** | **Dyrep** | **Roland** |   | **TGN** | **Jodie** | **Dyrep** | **Roland** |
> | Vanilla Acc. |       | 0.9         | 0.87  | 0.9   | 0.88   |   | 0.93         | 0.92  | 0.91  | 0.91   |
> | $t_{0}$     | FN    | 0.76        | 0.82  | 0.77  | 0.79   |   | 0.79         | 0.84  | 0.77  | 0.81   |
> |         | Meta-h  | 0.86        | 0.83  | 0.88  | 0.85   |   | 0.88         | 0.88  | 0.87  | 0.89   |
> |         | TDGIA | 0.72        | 0.81  | 0.74  | 0.76   |   | 0.76         | 0.81  | 0.75  | 0.76   |
> |         | ours  | 0.84        | 0.85  | 0.81  | 0.78   |   | 0.85         | 0.83  | 0.83  | 0.82   |
> | $t_{25}$     | FN    | 0.9         | 0.86  | 0.89  | 0.88   |   | 0.91         | 0.92  | 0.9   | 0.9    |
> |         | Meta-h  | 0.89        | 0.86  | 0.9   | 0.87   |   | 0.93         | 0.92  | 0.9   | 0.91   |
> |         | TDGIA | 0.89        | 0.85  | 0.89  | 0.88   |   | 0.91         | 0.92  | 0.89  | 0.89   |
> |         | ours  | 0.81        | 0.84  | 0.76  | 0.8    |   | 0.79         | 0.83  | 0.77  | 0.82   |
> | $t_{50}$    | FN    | 0.9         | 0.86  | 0.9   | 0.88   |   | 0.93         | 0.92  | 0.9   | 0.9    |
> |         | Meta-h  | 0.9         | 0.86  | 0.9   | 0.88   |   | 0.93         | 0.92  | 0.91  | 0.91   |
> |         | TDGIA | 0.89        | 0.86  | 0.9   | 0.87   |   | 0.92         | 0.92  | 0.9   | 0.9    |
> |         | ours  | 0.77        | 0.82  | 0.76  | 0.77   |   | 0.77         | 0.82  | 0.75  | 0.81   |
>
>
> We also added the results to **Table 1 in Section 5.2 and Table 3&5 in Appendix B.3** in our revised paper.

---

> ### Author Response · Authors · 2023-11-20
> **Author Response to Reviewer LiVK (3/4)**
>
> - #### **Different Attack budget and target node density:**
>
> Moreover, we evaluate how our approach works under different numbers of target nodes (i.e., attack budget) and densities of target nodes. Specifically, we investigate the edge prediction accuracies of our attack under different budgets and check if there is a difference between selecting 1% of highest-degree and lowest-degree nodes as victim nodes. We show the results in **Table R.7** and **Table R.8**. The results demonstrate that: (1) Our attack can be effective and scale under different attack budgets. (2) It is helpful to select high-degree victim nodes since lowest-degree nodes lead to worse performances.
>
> > **Table R.7.** Accuracies of our attack under different attack budgets
> |            |     | 1%   | 3%   | 5%   | 7%   | 9%   | 11%  | 13%  | 15%  |
> |------------|-----|------|------|------|------|------|------|------|------|
> | TGN+WIKI   | t0  | 0.89 | 0.84 | 0.81 | 0.82 | 0.74 | 0.77 | 0.76 | 0.71 |
> |            | t25 | 0.83 | 0.81 | 0.8  | 0.75 | 0.76 | 0.7  | 0.69 | 0.64 |
> |            | t50 | 0.85 | 0.8  | 0.75 | 0.72 | 0.73 | 0.7  | 0.67 | 0.63 |
> |            |     |      |      |      |      |      |      |      |      |
> | TGN+REDDIT | t0  | 0.93 | 0.88 | 0.75 | 0.78 | 0.72 | 0.78 | 0.68 | 0.72 |
> |            | t25 | 0.84 | 0.82 | 0.81 | 0.76 | 0.74 | 0.72 | 0.73 | 0.69 |
> |            | t50 | 0.83 | 0.8  | 0.8  | 0.74 | 0.75 | 0.71 | 0.69 | 0.68 |
>
> > **Table R.8.** Number of affected nodes and accumulated accuracies of our attack under selecting 1% highest/lowest-degree victim nodes.
> |                |                    | t0   | t10  | t20  | t30  | t40  | t50  |
> |----------------|--------------------|------|------|------|------|------|------|
> | **Highest Degree** | # of Affected Node | 74   | 176  | 211  | 253  | 285  | 313  |
> |                | Overall Acc.       | 0.91 | 0.88 | 0.86 | 0.86 | 0.85 | 0.85 |
> |                |                    |      |      |      |      |      |      |
> | **Lowest Degree**  | # of Affected Node | 79   | 134  | 159  | 174  | 192  | 211  |
> |                | Overall Acc.       | 0.83 | 0.88 | 0.87 | 0.87 | 0.87 | 0.86 |
>
> We added more results to **Figure 8 in Section 5.2 and Figure 15 in Appendix B.6** in our revised paper.
>
> [1] Zou, Xu, et al. "Tdgia: Effective injection attacks on graph neural networks." Proceedings of the 27th ACM SIGKDD Conference on Knowledge Discovery & Data Mining. 2021.
>
> [2] Li, Kuan, et al. "Revisiting graph adversarial attack and defense from a data distribution perspective." The Eleventh International Conference on Learning Representations. 2022.
>
> [3] You, Jiaxuan, Tianyu Du, and Jure Leskovec. "ROLAND: graph learning framework for dynamic graphs." Proceedings of the 28th ACM SIGKDD Conference on Knowledge Discovery and Data Mining. 2022.
>
> [4] S. Kumar, W. L. Hamilton, J. Leskovec, and D. Jurafsky. Community interaction and conflict on the web. In Proceedings of the 2018 World Wide Web Conference on World Wide Web, pages 933–943. International World Wide Web Conferences Steering Committee, 2018
>
>
>
> ## **Q2. Limitations of our attack under diverse input graphs and models**
>
>
> According to the performance variance between different models, we can observe that while our approach can effectively mislead TGN, ROLAND, and DyRep, its effectiveness is less significant on JODIE, which uses differences between a node's current and its last update time to decay the memory.
>
>
> From these observations, we deduce that our attack may encounter limitations in two specific scenarios:
> - **Limited Influence of Node Memory on Predictions:** Our attack's effectiveness may be mitigated in situations where the node memory has a relatively minor role in influencing the model's predictions.
> - **Usage of Additional Information in TGNN Models:** The effectiveness may also be constrained when the targeted TGNN model incorporates additional information beyond the node memory for its predictive processes.
> While our attack strategy outperforms the baselines, these insights highlight potential limitations under certain model-specific conditions.

---

> ### Author Response · Authors · 2023-11-20
> **Author Response to Reviewer LiVK (4/4)**
>
> ## **Q3. Potential defenses against the proposed method**
>
> Thank you for your valuable suggestion. To evaluate the effectiveness of our approach under various adversarial defense strategies, we have incorporated the following methods:
> - **Adversarial Training:** In line with the approach detailed in [5], we introduce perturbations to the node memories in TGNN models during the training. We then employ a minimax adversarial training scheme to enhance the robustness of the TGNN model against these perturbations.
> - **Regularization under Empirical Lipschitz Bound:** Following the methodology in [6], we apply a Lipschitz bound during the TGNN training process. This regularization aims to bound the effectiveness of small perturbations, such as adversarial examples.
>
>
> The results are shown in **Table R.9**. The results demonstrate that our attack can remain effective with TGNNs guarded by state-of-the-art defense schemes.
>
> >**Table R.9.** Accumulated accuracy over different timestamps of the vanilla/attacked TGNNs under different defenses
> |              | **Model**   |      | **TGN**  | **Jodie** | **Dyrep** | **Roland** |
> |--------------|---------|------|------|-------|-------|--------|
> | **Adversarial Training**  | Vanilla |      | 0.88 | 0.78  | 0.81  | 0.91   |
> |   | $t_{0}$      | ours | 0.81 | 0.75  | 0.73  | 0.84   |
> |              |         | FN   | 0.77 | 0.73  | 0.66  | 0.80   |
> |              | $t_{25}$     | ours | 0.82 | 0.75  | 0.72  | 0.82   |
> |              |         | FN   | 0.87 | 0.77  | 0.80  | 0.90   |
> |              | $t_{50}$    | ours | 0.80 | 0.76  | 0.70  | 0.80   |
> |              |         | FN   | 0.88 | 0.78  | 0.81  | 0.91   |
> |              |         |      |      |       |       |        |
> | **Regulation**   | Vanilla |      | 0.90 | 0.82  | 0.70  | 0.92   |
> |              | $t_{0}$      | ours | 0.87 | 0.75  | 0.58  | 0.86   |
> |              |         | FN   | 0.84 | 0.74  | 0.60  | 0.84   |
> |              | $t_{25}$     | ours | 0.81 | 0.73  | 0.47  | 0.83   |
> |              |         | FN   | 0.90 | 0.82  | 0.70  | 0.92   |
> |              | $t_{50}$     | ours | 0.81 | 0.76  | 0.48  | 0.81   |
> |              |         | FN   | 0.90 | 0.82  | 0.70  | 0.92   |
>
> We also added more results to **Figure 6 in Section 5.2 and Figure 12&13 in Appendix B.4** in our revised paper. We also include an introduction of these defenses in **Appendix B.3** in our revised paper for comprehensive understanding.
>
>
> Notably, most robust GCN models, such as RobustGCN, SGCN, GraphSAGE, and TAGCN mentioned in ROLAND [3], are primarily tailored for static graph benchmarks. Given their design constraints, these models are unsuited for TGNN setup with dynamic graph benchmarks and do not offer a viable defense for the TGNN models targeted by our attack.
>
>
> However, we expect a potential attack-oriented defense scheme for our attack using memory filtering. **Specifically, a potential defensive approach is to pay less attention to the nodes' memory and rely more on their current input adaptively.**
>
>
> This scheme stems from the observation that our attacks are less effective on JODIE. One key difference in JODIE is that it decays the node memory based on the time differences between the prediction time and the node's last update time. This mechanism introduces more hints (i.e., time differences) in addition to the memory itself, which cannot be effectively distorted by the attacks and yields some crucial information. For example, a Wikipedia user is less likely to be banned if he/she makes a new post after being inactive for a long while. Therefore, using this non-memory information or current information that does not interact with node memory could effectively hinder adversarial noises. To this end, an intelligent defense mechanism can judiciously filter out the memory and adaptively focus more on non-memory information if the memory is suspicious or potentially noisy. We also include these discussions in **Appendix C** in our revised paper.
>
>
> [5] Madry, Aleksander, et al. "Towards deep learning models resistant to adversarial attacks." arXiv preprint arXiv:1706.06083 (2017).
>
> [6] Jia, Yaning, et al. "Enhancing node-level adversarial defenses by Lipschitz regularization of graph neural networks." Proceedings of the 29th ACM SIGKDD Conference on Knowledge Discovery and Data Mining. 2023.

---

> ### Author Response · Authors · 2023-11-22
>
> Dear Reviewer LiVK,
>
> Thanks for your time and reviewing efforts! We appreciate your constructive comments and positive feedbacks.
>
> We provide suggested results and discussions in the authors' response, we hope our responses have answered your questions.  As the deadline for open discussion is soon, please don't hesitate to respond if you have any further concerns.
>
> Best,
>
> Authors

---

> > ### Comment · Reviewer_LiVK · 2023-12-01
> >
> > Thanks for the feedback. I do not have any concern currently.

---

### Official Review · Reviewer_aaPY · 2023-11-01

**Soundness:** 2 fair
**Presentation:** 2 fair
**Contribution:** 2 fair
**Rating:** 5
**Confidence:** 4

**Summary:**

This paper proposes a framework called MemStranding, which is used to attack Temporal Graph Neural Networks (TGNNs) by leveraging node memories to create adversarial noises in dynamic graphs. The authors provide experimental results to demonstrate the effectiveness of MemStranding in decreasing the performance of TGNN models in various tasks.

**Strengths:**

1. Attacking graph under GNNs is promising
2. The paper discusses real-world scenarios where dynamic graphs are prevalent, which highlights the relevance and importance of the proposed framework.
3. The authors identify the limitations of existing adversarial attacks on TGNNs and explore the challenges of adapting them to TGNNs within these constraints.

**Weaknesses:**

1. The experiments are weak
2. The presentations are unclear
3. The comparisons are weak

**Questions:**

1. The authors do not provide a comprehensive comparison of MemStranding with other existing adversarial attacks on TGNNs. For example, TIGIA [1] and a lot of methods in surveys [2] propose injective attacks. But the author only compares one the fakenode baseline.

2. The experimental results are limited to small datasets, which may not be sufficient to generalize the effectiveness of MemStranding in real-world applications.

3. It only uses limited TGNNs for evaluations. Recently, researchers have proposed more powerful TGNNs, such as roland [3]

4. It only uses raw TGNN for evaluations without using current GNN defenders for evaluations. Considering current platforms may use GNN defenders instead of raw GNNs, it should explore the effectiveness of attackers under GNN defenders.

5. The paper assumes that the attacker has complete knowledge of the graph structure and node attributes before each time t, which may not be realistic in real-world scenarios.

6. unclear part: Section 4.1 is unclear, which should add more explanations on why coverage state is so important.

[1]Zou, Xu, et al. "Tdgia: Effective injection attacks on graph neural networks." Proceedings of the 27th ACM SIGKDD Conference on Knowledge Discovery & Data Mining. 2021.

[2] Sun, Lichao, et al. "Adversarial attack and defense on graph data: A survey." IEEE Transactions on Knowledge and Data Engineering (2022).

[3] You, Jiaxuan, Tianyu Du, and Jure Leskovec. "ROLAND: graph learning framework for dynamic graphs." Proceedings of the 28th ACM SIGKDD Conference on Knowledge Discovery and Data Mining. 2022.

**Details Of Ethics Concerns:**

The paper focuses solely on attacking TGNN models and does not explore the potential defenses against such attacks.

---

> ### Author Response · Authors · 2023-11-20
> **Author Response to Reviewer aaPY (1/5)**
>
> ## **Comments:**
> We appreciate the valuable comments and suggestions from the reviewer. We add more results as suggested by the reviewer, including comparisons with more baseline attacks on more TGNN models and datasets, and evaluate the performance under various adversarial defenses. We clarify the questions and revise the paper carefully. We hope our response can help alleviate the reviewer's concern.
>
> ## **Q1~Q3. Comparisons with more baseline attacks on more TGNNs in larger datasets.**
>
> Thank you for the kind suggestion.
>
> - #### **Additional Baseline Attacks and TGNN model:**
>
> To provide more comprehensive comparisons, we compare our work with two more recent state-of-the-art attacks for static GNNs: **TDGIA**[1] and **Meta-Attack-Heuristic (Meta-h)** mentioned in [2]. Additionally, as suggested, we also include the evaluation of our work in conjunction with **ROLAND**[3]. We show the results under the 5% node attack budget in **Table R.1.** The results indicate that our attack can outperform various baselines on more recent TGNN models.
>
> >**Table R.1.:** Accumulated accuracy over different timestamps of the vanilla/attacked TGNNs in WIKI and REDDIT.
> | Dataset |       | Wiki |       |       |        |   | Reddit |       |       |        |
> |---------|-------|------|-------|-------|--------|---|--------|-------|-------|--------|
> | **Model**   |       | **TGN** | **Jodie** | **Dyrep** | **Roland** |   | **TGN** | **Jodie** | **Dyrep** | **Roland** |
> | Vanilla Acc. |       | 0.93 | 0.87  | 0.85  | 0.94   |   | 0.96   | 0.98  | 0.96  | 0.95   |
> | $t_0$| FN    | 0.81 | 0.74  | 0.74  | 0.82   |   | 0.84   | 0.83  | 0.84  | 0.83   |
> |         | Meta-h  | 0.9  | 0.83  | 0.81  | 0.85   |   | 0.93   | 0.95  | 0.9   | 0.92   |
> |         | TDGIA | 0.77 | 0.72  | 0.71  | 0.8    |   | 0.74   | 0.8   | 0.81  | 0.74   |
> |         | ours  | 0.89 | 0.88  | 0.85  | 0.87   |   | 0.75   | 0.84  | 0.94  | 0.82   |
> | $t_{25}$ | FN    | 0.92 | 0.87  | 0.85  | 0.94   |   | 0.97   | 0.97  | 0.96  | 0.93   |
> |         | Meta-h  | 0.93 | 0.87  | 0.84  | 0.93   |   | 0.96   | 0.98  | 0.94  | 0.96   |
> |         | TDGIA | 0.93 | 0.81  | 0.84  | 0.92   |   | 0.94   | 0.95  | 0.95  | 0.9    |
> |         | ours  | 0.8  | 0.8   | 0.78  | 0.85   |   | 0.81   | 0.84  | 0.91  | 0.8    |
> | $t_{50}$  | FN    | 0.94 | 0.87  | 0.86  | 0.94   |   | 0.97   | 0.97  | 0.96  | 0.95   |
> |         | Meta-h  | 0.93 | 0.87  | 0.85  | 0.93   |   | 0.97   | 0.98  | 0.94  | 0.95   |
> |         | TDGIA | 0.93 | 0.87  | 0.85  | 0.93   |   | 0.96   | 0.97  | 0.95  | 0.92   |
> |         | ours  | 0.75 | 0.81  | 0.76  | 0.84   |   | 0.8    | 0.84  | 0.91  | 0.8    |
>
>
> We also include more results in **Table 1 in Section 5.2 and Table 5&7 in Appendix B.3** in our revised paper.
>
> Additionally, we would like to mention that our approach is the first attack on TGNNs within the domain of Continuous-Time Dynamic Graphs (CTDG). Existing methodologies and benchmarks in the literature primarily cater to static graphs. Consequently, there are no established baseline approaches or precedents on CTDG-based TGNNs to compare our work against directly. Therefore, we leverage the core mechanisms of these attacks and implement their attacks in the dynamic graph scenarios. More detailed information regarding the baseline attack setups has been added to Appendix B.2 for comprehensive understanding.

---

> ### Author Response · Authors · 2023-11-20
> **Author Response to Reviewer aaPY (2/5)**
>
> - ####  **Additional Dataset:**
>
> To measure if our approach can generalize the larger real-world applications, we evaluate our works on two larger datasets from real-world social media: **Reddit-body** and **Reddit-title** [4].
> Compared to WIKI and REDDIT, which contain approximately 10,000 nodes, these datasets have more nodes (i.e., 35,000 and 54,000 nodes, respectively). The details of the datasets are introduced in **Table 2 in Appendix B.1** in our revised paper. We show the results under the 5% node attack budget in **Table R.2**. The results indicate that our attack can outperform baselines on larger graphs.
>
>
> >**Table R.2** Accumulated accuracy over different timestamps of the vanilla/attacked TGNNs in Reddit-body and Reddit-title.
> | **Dataset** |       | **Reddit-body** |       |       |        |   | **Reddit-title** |       |       |        |
> |---------|-------|-------------|-------|-------|--------|---|--------------|-------|-------|--------|
> | **Model**   |       | **TGN** | **Jodie** | **Dyrep** | **Roland** |   | **TGN** | **Jodie** | **Dyrep** | **Roland** |
> | Vanilla Acc. |       | 0.9         | 0.87  | 0.9   | 0.88   |   | 0.93         | 0.92  | 0.91  | 0.91   |
> | $t_{0}$     | FN    | 0.76        | 0.82  | 0.77  | 0.79   |   | 0.79         | 0.84  | 0.77  | 0.81   |
> |         | Meta-h  | 0.86        | 0.83  | 0.88  | 0.85   |   | 0.88         | 0.88  | 0.87  | 0.89   |
> |         | TDGIA | 0.72        | 0.81  | 0.74  | 0.76   |   | 0.76         | 0.81  | 0.75  | 0.76   |
> |         | ours  | 0.84        | 0.85  | 0.81  | 0.78   |   | 0.85         | 0.83  | 0.83  | 0.82   |
> | $t_{25}$     | FN    | 0.9         | 0.86  | 0.89  | 0.88   |   | 0.91         | 0.92  | 0.9   | 0.9    |
> |         | Meta-h  | 0.89        | 0.86  | 0.9   | 0.87   |   | 0.93         | 0.92  | 0.9   | 0.91   |
> |         | TDGIA | 0.89        | 0.85  | 0.89  | 0.88   |   | 0.91         | 0.92  | 0.89  | 0.89   |
> |         | ours  | 0.81        | 0.84  | 0.76  | 0.8    |   | 0.79         | 0.83  | 0.77  | 0.82   |
> | $t_{50}$    | FN    | 0.9         | 0.86  | 0.9   | 0.88   |   | 0.93         | 0.92  | 0.9   | 0.9    |
> |         | Meta-h  | 0.9         | 0.86  | 0.9   | 0.88   |   | 0.93         | 0.92  | 0.91  | 0.91   |
> |         | TDGIA | 0.89        | 0.86  | 0.9   | 0.87   |   | 0.92         | 0.92  | 0.9   | 0.9    |
> |         | ours  | 0.77        | 0.82  | 0.76  | 0.77   |   | 0.77         | 0.82  | 0.75  | 0.81   |
>
>
> We also added the results to **Table 1 in Section 5.2 and Table 3&5 in Appendix B.3** in our revised paper.
>
> [1] Zou, Xu, et al. "Tdgia: Effective injection attacks on graph neural networks." Proceedings of the 27th ACM SIGKDD Conference on Knowledge Discovery & Data Mining. 2021.
>
> [2] Li, Kuan, et al. "Revisiting graph adversarial attack and defense from a data distribution perspective." The Eleventh International Conference on Learning Representations. 2022.
>
> [3] You, Jiaxuan, Tianyu Du, and Jure Leskovec. "ROLAND: graph learning framework for dynamic graphs." Proceedings of the 28th ACM SIGKDD Conference on Knowledge Discovery and Data Mining. 2022.
>
> [4] S. Kumar, W. L. Hamilton, J. Leskovec, and D. Jurafsky. Community interaction and conflict on the web. In Proceedings of the 2018 World Wide Web Conference on World Wide Web, pages 933–943. International World Wide Web Conferences Steering Committee, 2018

---

> ### Author Response · Authors · 2023-11-20
> **Author Response to Reviewer aaPY (3/5)**
>
> ## **Q4. Evaluation results on adversarial defenses.**
> Thank you for your valuable suggestion. To evaluate the effectiveness of our approach under various adversarial defense strategies, we have incorporated the following methods:
> - **Adversarial Training:** In line with the approach detailed in [5], we introduce perturbations to the node memories in TGNN models during the training. We then employ a minimax adversarial training scheme to enhance the robustness of the TGNN model against these perturbations.
> - **Regularization under Empirical Lipschitz Bound:** Following the methodology in [6], we apply a Lipschitz bound during the TGNN training process. This regularization aims to bound the effectiveness of small perturbations, such as adversarial examples.
>
> The results are shown in **Table R.3**. The results demonstrate that our attack can remain effective with TGNNs guarded by state-of-the-art defense schemes.
>
> >**Table R.3.** Accumulated accuracy over different timestamps of the vanilla/attacked TGNNs under different defenses
> |              | **Model**   |      | **TGN**  | **Jodie** | **Dyrep** | **Roland** |
> |--------------|---------|------|------|-------|-------|--------|
> | **Adversarial Training**  | Vanilla |      | 0.88 | 0.78  | 0.81  | 0.91   |
> |   | $t_{0}$      | ours | 0.81 | 0.75  | 0.73  | 0.84   |
> |              |         | FN   | 0.77 | 0.73  | 0.66  | 0.80   |
> |              | $t_{25}$     | ours | 0.82 | 0.75  | 0.72  | 0.82   |
> |              |         | FN   | 0.87 | 0.77  | 0.80  | 0.90   |
> |              | $t_{50}$    | ours | 0.80 | 0.76  | 0.70  | 0.80   |
> |              |         | FN   | 0.88 | 0.78  | 0.81  | 0.91   |
> |              |         |      |      |       |       |        |
> | **Regulation**   | Vanilla |      | 0.90 | 0.82  | 0.70  | 0.92   |
> |              | $t_{0}$      | ours | 0.87 | 0.75  | 0.58  | 0.86   |
> |              |         | FN   | 0.84 | 0.74  | 0.60  | 0.84   |
> |              | $t_{25}$     | ours | 0.81 | 0.73  | 0.47  | 0.83   |
> |              |         | FN   | 0.90 | 0.82  | 0.70  | 0.92   |
> |              | $t_{50}$     | ours | 0.81 | 0.76  | 0.48  | 0.81   |
> |              |         | FN   | 0.90 | 0.82  | 0.70  | 0.92   |
>
> We also added more results to **Figure 6 in Section 5.2 and Figure 12&13 in Appendix B.4** in our revised paper. We also include an introduction of these defenses in **Appendix B.3** in our revised paper for comprehensive understanding.
>
> Notably, most robust GCN models, such as RobustGCN, SGCN, GraphSAGE, and TAGCN mentioned in ROLAND [3], are primarily tailored for static graph benchmarks. Given their design constraints, these models are unsuited for TGNN setup with dynamic graph benchmarks and do not offer a viable defense for the TGNN models targeted by our attack.
>
> [5] Madry, Aleksander, et al. "Towards deep learning models resistant to adversarial attacks." arXiv preprint arXiv:1706.06083 (2017).
>
> [6] Jia, Yaning, et al. "Enhancing node-level adversarial defenses by Lipschitz regularization of graph neural networks." Proceedings of the 29th ACM SIGKDD Conference on Knowledge Discovery and Data Mining. 2023.

---

> ### Author Response · Authors · 2023-11-20
> **Author Response to Reviewer aaPY (4/5)**
>
> ## **Q5. Why assume the attacker has complete knowledge of the graph structure and node attributes before each time t?**
> Thank you for your insightful question. Our assumption is grounded in practicality for two key reasons:
> - **Observable Historical Graph Structure:** In real-world dynamic graphs, the historical graph structure is typically accessible. For example, data on platforms like Reddit and Wikipedia allows public access, and one can systematically gather all historical information related to targeted posts (i.e., a node in the Reddit graph).
> - **Publicly Accessible Message Embedding Techniques:** Message information is embedded using publicly available methods, such as LIWC [8] or BERT [9]. Additionally, in TGNN prediction frameworks, the node features attributes are initiated from an all-zero state. Thus, by collating messages related to a specific node (like comments linked to a post), we can effortlessly reconstruct the attributes of the target node.
>
>
> Our assumption is aligned with the assumption for attacks in static GNN (our baseline methodologies, i.e., FakeNode[7], TDGIA[1], and Meta-Attack-Heuristic (Meta-h) mentioned in [2]). These attacks uniformly assume that the attacker has complete knowledge of the input graphs, including graph structure and node attributes. Significantly, our method is designed to require only historical information related to the target nodes and, therefore, demands less information compared to the assumptions in previous studies.
>
>
> We concur that these attacks are less effective when the graph structure and node information are entirely obscured from public view. In scenarios where the data is completely private, we need to adopt system-side attack methodologies [10] to access private data before deploying these attacks. However, such considerations fall outside the scope of both our study and those of our baseline references.
>
> [7] Wang, Xiaoyun, et al. "Fake node attacks on graph convolutional networks." Journal of Computational and Cognitive Engineering 1.4 (2022): 165-173.
>
> [8] Pennebaker, James W., Martha E. Francis, and Roger J. Booth. "Linguistic inquiry and word count: LIWC 2001." Mahway: Lawrence Erlbaum Associates 71.2001 (2001): 2001.
>
> [9] Devlin, Jacob, et al. "Bert: Pre-training of deep bidirectional transformers for language understanding." arXiv preprint arXiv:1810.04805 (2018).
>
> [10] Chandra, Sudipta, Soumya Ray, and R. T. Goswami. "Big data security: survey on frameworks and algorithms." 2017 IEEE 7th International Advance Computing Conference (IACC). IEEE, 2017.

---

> ### Author Response · Authors · 2023-11-20
> **Author Response to Reviewer aaPY (5/5)**
>
> ## **Q6. Why is coverage state so important?**
> Thank you for pointing out the need for clarity in **Section 4.1** regarding the importance of a node's converge state. The converge state of a node plays a pivotal role in our analysis, primarily due to its impact on the stability of the node's memory. **Figure 4** of our paper illustrates that a node surrounded by similar neighbors tends to fall into a stable converge state after a few updates. The interactions or 'messages' from these neighboring nodes effectively push the node's memory towards this state, subsequently maintaining its stability.
>
>
> It's important to highlight that although having similar neighbors is necessary for stabilizing the node's memory, it is insufficient. The memory of a node can only achieve stabilization upon reaching its converge state. Therefore, it is significant to forge the node’s noisy memory into the converged state to prevent this converging process from breaking our noise. In other words, if we do not produce a noisy and converged memory state for a target node, the injected noise could be lost during its converging process.
>
>
> To demonstrate the necessity of the converged state, we show the effectiveness of our attack without ensuring this state. Specifically, we show the similarities between root nodes' initial noisy memories (at the time of the attack) and themselves'/their subsequent neighbors' memories in MemSranding with and w/o converge state in **Table R.4**. If the converge states are not guaranteed, the similarities also suffer drops and achieve much lower similarities in the future. This is because the memories will change before they reach their converged states, making the final converged state different from the original noise ones, indicating that the converged state is essential for persisting noisy memories.
>
>
> >**Table R.4.** The similarities between root nodes' initial noisy memories (at the time of the attack) and themselves'/their subsequent neighbors' memories in MemSranding with and w/o converge state.
> |                     |                  | $t_{1}$   | $t_{3}$   | $t_{5}$   | $t_{7}$   | $t_{9}$   | $t_{10}$  | $t_{11}$  | $t_{13}$  |
> |---------------------|------------------|------|------|------|------|------|------|------|------|
> | **With Converge State** | Root Node        | 0.98 | 0.97 | 0.96 | 0.96 | 0.96 | 0.95 | 0.95 | 0.94 |
> |                     | One-Hop Neighber | 0.52 | 0.76 | 0.84 | 0.88 | 0.90 | 0.91 | 0.92 | 0.93 |
> |                     | Two-Hop Neighber | 0.26 | 0.54 | 0.68 | 0.79 | 0.83 | 0.86 | 0.89 | 0.90 |
> | **W/O Converge State**  | Root Node        | 0.96 | 0.90 | 0.89 | 0.84 | 0.84 | 0.81 | 0.79 | 0.78 |
> |                     | One-Hop Neighber | 0.25 | 0.43 | 0.58 | 0.58 | 0.67 | 0.69 | 0.61 | 0.66 |
> |                     | Two-Hop Neighber | 0.02 | 0.27 | 0.46 | 0.57 | 0.59 | 0.60 | 0.58 | 0.63 |
>
>
> We also include more results in **Figure 7 in Section 5.2, and Figure 14 in Appendix B.5** in our revised paper. Additionally, we revise our description in **Section 4.1** to emphasize the importance of reaching the converged state.

---

> ### Author Response · Authors · 2023-11-22
> **Looking forward to your feedback**
>
> Dear Reviewer aaPY,
>
> We would like to sincerely thank you again for your thoughtful suggestions to improve our work.
>
> We provide additional experimental results and explanations in the authors' response and revision. As the deadline for open discussion is soon, we would sincerely hope to use this opportunity to see if our responses are sufficient and if any concern remains. It will be our great pleasure if you would consider updating your review or score.
>
> Thanks again for your time.
>
> Best,
>
> Authors

---

> > ### Comment · Reviewer_aaPY · 2023-11-22
> > **Rebuttal Reply**
> >
> > Thanks for your new results and explanations. Currently, I think this new stuff has addressed some of my concerns, but the defense GNN model used for evaluation is still limited.
> >
> > Also, considering the real datasets have millions of nodes, this paper only uses graphs with ~50 thousand nodes, which is still limited.
> >
> > Based on this rebuttal, I will increase the score accordingly.

---

> > > ### Author Response · Authors · 2023-11-23
> > > **Follow-up Response (2/2)**
> > >
> > > - #### **For larger datasets.**
> > >
> > > To measure our approach on a larger dataset, we select the largest temporal graph dataset on the SNAP datasetcollection[10]---wiki-talk-temporal[11]—for further analysis. This dataset represents Wikipedia users editing each other's Talk page. A directed edge ($u$, $v$, $t$) means user $u$ edited $v$ 's talk page at time $t$. **The graph has 1,140,149 nodes, and 7,833,140 collected over 2320 days.**
> > >
> > >
> > > The dataset has non-attributed edges, so we set them as all zero vectors. Note that we set the memory size to 64 instead of 172 to avoid the Out-Of-Memory issue. Due to the time limit, we train TGN and Roland for 10 epochs instead of 20 in our prior experimental settings. The results are shown in **Table F.2**. As we can observe, even for a very large graph with a 1% node budget, our attack shows a similar behavior as our prior results --Our attack is long-lasting and can affect more nodes' predictions in the future.
> > >
> > > >**Table F.2.** Attack Perfromance on wiki-talk-temporal
> > > | **Atack Budget** |          | **1%**   |        |   | **5%**   |        |
> > > |--------------|----------|------|--------|---|------|--------|
> > > | **Model**        |          | **TGN**  | **Roland** |   | **TGN**  | **Roland** |
> > > | Vanilla Acc. |          | 0.97 | 0.98   |   | 0.97 | 0.98   |
> > > | $t_{0}$           | Ours     | 0.94 | 0.91   |   | 0.86 | 0.88   |
> > > |              | FakeNode | 0.89 | 0.9    |   | 0.83 | 0.88   |
> > > | $t_{25}$           | Ours     | 0.92 | 0.9    |   | 0.82 | 0.89   |
> > > |              | FakeNode | 0.98 | 0.97   |   | 0.97 | 0.96   |
> > > | $t_{50}$           | Ours     | 0.91 | 0.91   |   | 0.84 | 0.86   |
> > > |              | FakeNode | 0.97 | 0.98   |   | 0.97 | 0.97   |
> > >
> > > [1] Zou, Xu, et al. "Tdgia: Effective injection attacks on graph neural networks." Proceedings of the 27th ACM SIGKDD Conference on Knowledge Discovery & Data Mining. 2021.
> > >
> > > [2] Thomas N Kipf and Max Welling. 2016. Semi-supervised classification with graph convolutional networks. arXiv preprint arXiv:1609.02907 (2016).
> > >
> > > [3] Will Hamilton, Zhitao Ying, and Jure Leskovec. 2017. Inductive representation learning on large graphs. In NeurIPS’17. 1024–1034.
> > >
> > > [4] Dingyuan Zhu, Ziwei Zhang, Peng Cui, and Wenwu Zhu. 2019. Robust graph convolutional networks against adversarial attacks. In KDD’19. 1399–1407
> > >
> > > [5] Felix Wu, Amauri Souza, Tianyi Zhang, Christopher Fifty, Tao Yu, and Kilian Weinberger. 2019. Simplifying graph convolutional networks. In ICML’19. PMLR, 6861–6871.
> > >
> > > [6] Keyulu Xu, Weihua Hu, Jure Leskovec, and Stefanie Jegelka. 2018. How Powerful are Graph Neural Networks? In ICLR’18.
> > >
> > > [7] Entezari, Negin, et al. "All you need is low (rank) defending against adversarial attacks on graphs." Proceedings of the 13th International Conference on Web Search and Data Mining. 2020.
> > >
> > > [8] Miller, Benjamin A., et al. "Improving robustness to attacks against vertex classification." MLG Workshop. 2019.
> > >
> > > [9] Zhang, Xiang, and Marinka Zitnik. "Gnnguard: Defending graph neural networks against adversarial attacks." Advances in neural information processing systems 33 (2020): 9263-9275.
> > >
> > > [10] Leskovec, Jure, and Rok Sosič. "Snap: A general-purpose network analysis and graph-mining library." ACM Transactions on Intelligent Systems and Technology (TIST) 8.1 (2016): 1-20.
> > >
> > > [11] Ashwin Paranjape, Austin R. Benson, and Jure Leskovec. "Motifs in Temporal Networks." In Proceedings of the Tenth ACM International Conference on Web Search and Data Mining, 2017.
> > >
> > >
> > > We sincerely appreciate your time. Hope our follow-up responses can address your concerns.

---

> ### Author Response · Authors · 2023-11-23
> **Follow-up Responses (1/2)**
>
> ## Followup: Comparisons with more defending methods and larger datasets.
>
> We would like to thank you again for your valuable time and suggestions for our responses.
>
> - #### **For limited adversarial defenses.**
>
> We agree that it is significant to investigate how our attack performs under diverse defending schemes. However, we would like to mention that many defenses for static GNNs/GCNs are not suitable/applicable to our attack scenarios. The main reason is that many adversarial defenses are specialized for static GNN/GCNs. As we mentioned in our earlier responses, many robust GNN models are especially well-tuned and specialized for static GNN setups. For example, TDGIA[1] adopts several robust GNNs that are either fine-tuned for the KDD-CUP or specially designed on top of static GCNs, such as layer-norm-GCN[2], GraphSage[3], RobustGCN[4], SGCN[5], GIN[6], etc. On the one hand, these defending methods rely on special static GCN model architectures (e.g., SGCN, RobustGCN) or static GNN models (e.g., GraphSage, GIN); on the other hand, these special GCN models are not transferable to TGNN tasks.
>
>
> We also consider some preprocessing defending schemes such as low-rank approximating[7] or data-filtering[8]. However, they are not natively suitable for TGNNs as well. The low-rank approximation [7] assumes that the graphs’ adjacency matrix is directly involved in the computing (like the GCN formula Y=AXW). It adopts Principal Component Analysis (PCA) on the adjacency matrix to remove redundant and non-robust information. However, in TGNN, the dynamic graphs are not represented in the adjacency matrix format but in a sequence of attributed events. Additionally, the adjacency matrix is not directly involved in the TGNN computing. The data-filtering[8][9] targets poison attacks (i.e., perturb training data to break model parameters). However, similar to TDGIA, our attack is an evasion attack, which misleads models by perturbing the input after the training.
>
>
> To this end, we chose adversarial training and regularization as our defense methods. Compared to the abovementioned methods, these defenses are more commonly adopted and transferable to TGNNs.
>
>
> To give a more in-depth evaluation, we design a defending method by leveraging the data-filtering concept in GNNGuard[9] for the evasion attack. Specifically, following the insights that only the similar node may provide significant information for prediction, GNNGuard[9] adopts a cosine-similarity-based approach to discount the messages passing between dissimilar nodes. So, we also use the cosine similarities to rank and filter the messages. Specifically, similar to the GNNGuard, we compute the similarities between two nodes. For each node, we normalize the similarities between it and its neighbors, then prune the lower 50% (same as GNNGuard). We show the experiment results in **Table F.1.**
>
> >**Table F.1.** Attack Performance under the GNNGurad
> | **Dataset**  |                    |          | **Wiki** |        | **Reddit** |        |
> |----|--------------------|----------|------|--------|--------|--------|
> | **Model**   |                    |          | **TGN**  | **Roland** | **TGN**    | **Roland** |
> |    Vanilla Acc. |       |          | 0.93 | 0.94   | 0.96   | 0.95   |
> |   After defense Acc. |  |          | 0.92 | 0.91   | 0.94   | 0.9    |
> | | | | | | |
> | 1% | $t_{0}$                 | Ours     | 0.87 | 0.88   | 0.91   | 0.88   |
> |    |                    | FakeNode | 0.87 | 0.81   | 0.9    | 0.84   |
> |    | $t_{25}$                | Ours     | 0.84 | 0.87   | 0.82   | 0.81   |
> |    |                    | FakeNode | 0.9  | 0.91   | 0.93   | 0.9    |
> |    | $t_{50}$                | Ours     | 0.83 | 0.85   | 0.81   | 0.81   |
> |    |                    | FakeNode | 0.92 | 0.91   | 0.94   | 0.9    |
> | | | | | | |
> | 5% | $t_{0}$                 | Ours     | 0.85 | 0.83   | 0.8    | 0.82   |
> |    |                    | FakeNode | 0.82 | 0.86   | 0.82   | 0.81   |
> |    | $t_{25}$               | Ours     | 0.79 | 0.81   | 0.81   | 0.8    |
> |    |                    | FakeNode | 0.91 | 0.91   | 0.92   | 0.9    |
> |    | $t_{50}$                | Ours     | 0.76 | 0.82   | 0.8    | 0.83   |
> |    |                    | FakeNode | 0.92 | 0.91   | 0.93   | 0.9    |

---

### Meta-Review · Area_Chair_xFPt · 2023-12-05

**Metareview:**

This paper focuses on the problem of adversarial attack on temporal graph neural networks. Two challenges (e.g., Noise Decaying and Knowledge Missing) in attacking temporal GNNs are first proposed. To address these challenges, the authors proposes an effective adversarial attack framework called MemStranding. Experimental results show that MemStranding can significantly decrease the TGNN models’ performances in various tasks. Specifically, the strength of this paper includes several aspects. 1) The paper focuses on adversarial attacks on TGNNs, studying the robustness of TGNNs to benefit their applications. 2) The authors provide a clear explanation of why typical attacks fail in dynamic graph scenarios. 3) In response to the failure of typical attacks, two attack goals are proposed: noise persisting and noise propagating, effectively addressing the issue.

However, there are several points to be further improved. For example, the defense GNN model used for evaluation is still limited. Also, considering the real datasets have millions of nodes, this paper only uses graphs with ~50 thousand nodes, which is still limited. Moreover, this paper lacks a comprehensive discussion of the limitations of the proposed method, including performance variations with different TGNN architectures and graph data characteristics. Therefore, this paper cannot be accepted at ICLR this time, but the enhanced version is highly encouraged to submit other top-tier venues.

**Justification For Why Not Higher Score:**

However, there are several points to be further improved. For example, the defense GNN model used for evaluation is still limited. Also, considering the real datasets have millions of nodes, this paper only uses graphs with ~50 thousand nodes, which is still limited. Moreover, this paper lacks a comprehensive discussion of the limitations of the proposed method, including performance variations with different TGNN architectures and graph data characteristics. Therefore, this paper cannot be accepted at ICLR this time, but the enhanced version is highly encouraged to submit other top-tier venues.

**Justification For Why Not Lower Score:**

However, there are several points to be further improved. For example, the defense GNN model used for evaluation is still limited. Also, considering the real datasets have millions of nodes, this paper only uses graphs with ~50 thousand nodes, which is still limited. Moreover, this paper lacks a comprehensive discussion of the limitations of the proposed method, including performance variations with different TGNN architectures and graph data characteristics. Therefore, this paper cannot be accepted at ICLR this time, but the enhanced version is highly encouraged to submit other top-tier venues.

---

### Decision · Program_Chairs · 2024-01-16

Reject